# Energy-Based Design: Improving Modern Brazilian Buildings Performance through Their Shading Systems, the Nova Cintra Case Study

Daniel Mateus [1,*] and Gonçalo Castro Henriques [2]

1   CIAUD, Research Centre for Architecture, Urbanism and Design, Lisbon School of Architecture, Universidade de Lisboa, 1349-063 Lisboa, Portugal
2   Faculdade de Arquitetura e Urbanismo, Universidade Federal do Rio de Janeiro, LAMO-PROURB, Rio de Janeiro 21941-901, Brazil; gch@fau.ufrj.br
*   Correspondence: dmateus@fa.ulisboa.pt or mateus.dani@gmail.com

**Abstract:** Current research applies an energy-based design model to improve performance in existing modern buildings, in Rio de Janeiro, from the 1940's, improving these buildings' shading systems. This article proposes a methodology tested through a case study, the Nova Cintra building. The methodology starts by analysing the original shading system performance, regarding insolation, illuminance and air temperature. Using these results, proposes two computacional methods to improve performance: (1) a combinatorial modelling process, recombining the existing shading systems positions in the building's north façade; and (2) a transformation process, using parametric and algorithmic–parametric modelling, to improve the existing shading systems performance. Both processes use optimization algorithms. The results of these modelling and optimization methods are compared with the results of the original system and suggests an improvement between 111.1% and 590.4% for insolation; between 360.9% and 84.4% for illuminance; and between 2.9% and 3.0% for air temperature, considering winter and summer solstices. This improvement aims at reducing the buildings' energy consumption and foresees the production of renewable energy from solar harvesting, to mitigate climate change.

**Keywords:** energy-based design; building performance; shading systems; carioca modern façades; insolation; illuminance; air temperature; computational methods; combinatorial, parametric and algorithmic–parametric modelling; optimization





## 1. Introduction

This article addresses the analysis and improvement of buildings' energy performance, through computational methods such as performance simulation, combinatorial modelling, parametric modelling and algorithmic–parametric modelling, used jointly with optimization algorithms.

It describes an ongoing research project, entitled Carioca Modern Façades, that studies the shading systems of a set of Brazilian modern buildings, from the carioca modernist school, from Rio de Janeiro. These buildings are from the 1940/50's, prior to the use of air-conditioning systems, and present original shading systems that are efficient in controlling excessive interior daylight, and also the overheating temperature in the building's interior space. This research project intends to analyse and improve the performance of the buildings by improving the existing shading systems. The present article describes the case study realized for one of these selected buildings, the Nova Cintra. It approaches in particular the Nova Cintra's north façade shading systems, the façade more exposed to the sun along the year.

The shading system design is critical in the building envelope design. The research uses the concept of building envelope proposed by Sadineni, Madala and Boehm [1], as a

key factor that determines the quality and controls the indoor conditions inside a building, irrespective of transient outdoor conditions. The building envelope behaves similarly to the human skin, as a barrier between the external environment and the organism. In fact, in the literature, the words skin or envelope are often used [2]. The shading systems have the capacity to filter the entry of natural daylight inside a building, which determines the illuminance (amount of natural light) and the air temperature of the building's interior space. Having a major impact on the energy consumption of the building. Hence, they are a crucial element of the building envelope. Beyond their direct influence in the energy consumption of a building, the shading systems also have the potential for renewable solar energy harvesting and for the production of electric energy, through the conversion of solar energy by the use of photovoltaic technology [3–5]. This fact makes shading systems elements that can improve the energy performance of a building, contributing also positively to mitigate climate change, diminishing the need of energy consumption from pollutant sources and increasing the energy production from renewable sources.

### 1.1. Literature Review

The literature review considers previous studies that use computational methods for analysis and to improve buildings' energy performance, considering the building envelope and/or the shading systems. References include Caldas and Norford's research [6] that proposes a design tool based on genetic algorithms and a design optimization method, to search the best design solutions regarding the windows of an office building, determining their position and size. The design optimizes the building's thermal and illuminance properties. Venâncio [7] develops and tests a parametric model for shading devices in early design stages, adopting multi-criteria analysis to optimize solutions to assure shading, blocking the incident solar radiation in the interior of rooms, and providing simultaneously adequate daylight levels. Jalali, Noorzai and Heidari [8] optimize an office building façade using genetic algorithms and multi-objective optimization, namely the Strength Pareto Evolutionary Algorithm (SPEA-2), to find a parametric set of optimal solutions regarding usable space, reduced thermal load and improved building natural daylight. Several other researchers use parametric modelling, performance analysis and optimization algorithms to find optimized design solutions for buildings [9–19]. Henriques, Duarte and Leal [20] develop an alternative strategy to find design solutions for a responsive skylight system with adequate illuminance performance, in a parametric solutions space. Instead of optimization algorithms, they use heuristics (sets of rules and methods to drive the discovery, invention and resolution of problems) to save time and resources to achieve good solutions. It is also worth mentioning the research by Vazquez, Duarte and Poerschke [21] that presents a digital framework to design masonry screen walls with optimized performance, associated with vernacular construction rules. Vazquez's research uses a generative design system based on shape grammars [22] and on existing construction rules. That design system is translated into a parametric model, connecting the system to a simulation engine to calculate the daylight results and cooling energy loads. Finally, Vazquez uses genetic algorithms to find families of optimized solutions, with visual feedback, to understand the trade-offs between such solutions. Granadeiro [23] develops a similar approach, realizing a shape grammar for Frank Lloyd Wright's prairie houses, converted to a parametric design system. Granadeiro connects this parametric design system with an energy simulation software, using genetic algorithms to find optimized building envelopes design solutions for the prairie houses, regarding energy consumption, in the parametric space of solutions generated through the prairie houses-shape grammar.

The research described in this article, as the researches mentioned above, uses parametric modelling/design with optimization algorithms as well to improve the energy performance of a building and its shading systems.

Parametric design models digital objects, such as buildings and their construction elements. Oxman [24] defines parametric design as an act of design thinking based on the process of exploring and re-editing associative relationships, in a geometrical solution

space. Parametric design is thus associated with the definition of a mathematical model. Accordingly, Caetano, Santos and Leitão [25] define parametric design as an approach that describes a design symbolically, using parameters. Therefore, parametric design uses a defined set of parameters to establish the shape of a model, or object, from the possible shapes that it can assume in a universe of solutions. This research defines a set of parameters for an existing shading system and develops a parametric modelling process to improve the shading system's performance. Here, we understand the term parametric modelling as a modelling process that stimulates parametric design thinking, providing a multiplicity of improved solutions.

Parametric modelling has the advantage of enabling quick exploration of a universe of shape solutions that share a set of common features, defined by a correspondent set of parameters. It presents, however, the disadvantage that the universe of shape solutions is limited, being the limitation larger or smaller depending on the number and type of parameters that define the solution universe. Thus, parametric modelling is restricted to the number of shapes that compose the universe of solutions. This restriction is a gap in the parametric modelling type.

Regarding optimization algorithms, genetic algorithms are used in the previously mentioned research. Caldas and Norford [6] highlight that a genetic algorithm is a procedure loosely based on the Darwinian notions of survival of the fittest, which uses selection and recombination operators to search, among candidate solutions, high-performance solutions. Genetic algorithms search for and evaluate design solutions, in a search guided by the results evaluation. The designer interacts with this loop process. According to Touloupaki and Theodosiou [19], genetic algorithms dominate the field of building design optimization regarding building envelopes [26], overall building form [27,28], HVAC [29] and renewable energy system [12] optimization. This research uses genetic algorithms as well, jointly with parametric modelling and algorithmic modelling, to find shading systems that have optimized energy performance.

### 1.2. Research Objectives

This research, regarding the identified parametric modelling gap, intends on the following:

1.  Develop and apply different modelling types for the shading systems of the Nova Cintra building's north façade that complement the parametric universe of shape solutions for the shading systems and that improve the energy performance;
2.  Implement an energy-based design process, proposed and developed in previous research [30,31], applying shape transformation processes on the Nova Cintra building's north façade shading systems, to improve energy performance. Shape transformations use different modelling processes, some addressed in the original energy-based design process and others developed and incorporated through this research.

The research associates combinatorial modelling with algorithmic–parametric modelling, to overcome parametric modelling limitations, increasing the universe of modelled solutions, thus increasing the possibility of improving energy performance.

A previous research stage developed combinatorial modelling [32], combining a set of different objects with a set of different positions. The objects combined are different shading systems combined in a set of positions, defined by the modern modular structure of the building, to attain the best performance regarding multi-objective goals.

Algorithmic–parametric modelling is a mixed process. It uses algorithmic modelling, which is the act of modelling objects through an algorithmic design process. Algorithmic design, according with Caetano, Santos and Leitão [25], is a design process based on algorithms. The Cambridge Dictionary [33] defines an algorithm as a "set of mathematical instructions or rules that, especially if given to a computer, will help to calculate an answer to a problem". Oxman [25] considers algorithmic design as the coding of explicit instructions to generate digital forms, while Queiroz and Vaz [34] state that algorithmic design allows the user to design directly through code manipulation. Similarly to the

parametric process in this research, algorithmic modelling is a modelling process that acts as an algorithmic design process. Algorithmic modelling creates shape variations through transformation operations, defined previously in the shape grammar entitled Building Envelopes Grammar [30,31]. The Building Envelopes Grammar operations can combine different arrangements and sequences, composing different algorithms that result in different shapes with improved energy performance, specifically regarding the harvesting of solar energy, to convert it to electric energy using photovoltaic technology. Therefore, this research uses Building Envelopes Grammar as an algorithmic modelling process, to transform the shading system's shape, improving its energetic performance. This research complements algorithmic modelling with parametric modelling, resulting in a mixed algorithmic–parametric process. The parametric modelling part of the mixed process uses the same parameters previously defined for the shading system, in the aforementioned initial single parametric modelling.

Combinatorial modelling and algorithmic–parametric modelling have different advantages and disadvantages. Combinatorial modelling has the advantage of permitting a very quick exploration of a set of previously defined shape solutions, in a set of different positions, to identify which solutions have the best performance for each position. However, it has the disadvantage of using a more limited set of shape solutions for each position.

Algorithmic–parametric modelling presents the point advantages of two modelling types. Algorithmic modelling allows fast shape creation or transformation operation that improves the performance of the modelled object. Parametric modelling allows complementing the algorithmic modelling process, exploring a set of solutions defined by parameters related with the shape solution created by the algorithmic process. The parametric exploration of solutions allows for further improving the performance of the modelled object. The algorithmic–parametric modelling process presents the overall advantage of exploring a wider universe of shape solutions with improved performance, having a higher probability of achieving the highest performance. Algorithmic–parametric modelling, however, has the disadvantage of slowing down regarding the combinatorial and parametric modelling types, to improve performance.

Comparing the three modelling types, algorithmic–parametric modelling obtains the largest universe of solutions, which leads to higher performance results. Parametric modelling allows for achieving the second largest universe of solutions. Combinatorial modelling presents the most restrict universe of solutions. Regarding the speed of calculation, combinatorial is the fastest, followed by the parametric and the algorithmic–parametric modelling types.

This energy-based design research uses parametric, combinatorial and algorithmic–parametric modelling types, together with optimization algorithms. The research improves the former energy-based design model [30,31], with parametric and combinatorial modelling types and through the junction of a parametric modelling process with the previously developed algorithmic modelling process. The use of optimization algorithms is also an improvement for the energy-based design model. The performances analysed and improved concern insolation (solar energy harvesting for conversion to electric energy), illuminance (quantity of natural illumination in the interior spaces of the building) and air temperature, with the main goal of obtaining buildings that minimize energy consumption and maximize energy production through renewable sources such as the Sun, enabling energy production for their own use.

*1.3. Research Stages*

The present article describes an ongoing research project, entitled Carioca Modern Façades, applying the energy-based design model on the shading systems of the selected set of modern Brazilian buildings, using as case study the Nova Cintra building.

The research proceeds in two stages. The first stage starts by analysing the Nova Cintra building. General analyses consider insolation (incident solar radiation) of the overall building envelope and illuminance analysis of the interior rooms of the building's

north façade. Followed the detailed insolation and illuminance analyses of the north façade shading systems, the research uses combinatorial modelling and multi-objective optimization, regarding insolation and illuminance, to generate alternative solutions for the shading systems.

The second research stage selected the upper left corner module of the north façade to generate alternative solutions improving the shading systems performance. This selection intends to shorten the analysis and generation process of optimized shading systems solutions by up to 3 months. This stage also analyses the air temperature in the room near the selected façade module. The research developed parametric modelling and algorithmic–parametric modelling, together with single-objective and multi-objective optimization, to generate alternative solutions for the shading systems in the selected façade module. The results obtained generated alternative shading solutions to improve performance.

The developments in the energy-based design model achieved with the realization of the first and second research stages intend to allow their appliance in a larger scale, to design shading systems of other new and refurbished buildings.

## 2. Materials and Methods

This section describes the research methodology. It starts by revealing the research framework, establishing a relation between the Carioca Modern Façades research project and the energy-based design model. This section describes the selected set of eight buildings for the Carioca Modern Façades and their façade characteristics as well.

Then, the research details the selected case study, the Nova Cintra building. After this, the methodology for the first research stage is revealed, which consisted in the 3D modelling of Nova Cintra, in the energy analysis of the north façade, considering insolation and illuminance, and in the combinatorial modelling and multi-objective optimization for the shading systems, regarding energy performance improvement through insolation and illuminance. Later, the research details the methodology for the second stage of the research, namely the development of the parametric and the algorithmic–parametric modelling processes, together with single-objective and multi-objective optimization processes, to improve energy performance for the Nova Cintra building.

The methodology of this research is replicable to the other buildings of the Carioca Modern Façades research project. Our methodology is also flexible, and might be used to analyse and improve all the building types that contain shading systems.

### 2.1. Research Framework: Carioca Modern Façades, an Energy-Based Design Model

This article is part of a wider research entitled Carioca Modern Façades. The main goal of the Carioca Modern Façades is to analyse and improve the energy performance of a set of eight buildings from the carioca modernist school, from Rio de Janeiro, Brazil. Following an energy-based design model, which constitutes a design model that uses shape generation and transformation processes, codified in shape grammars, to improve buildings' energy performance. Below (Figure 1) are displayed the selected eight buildings for the research.

These buildings of the carioca modern building heritage use the façade as an architectural filter, relating interior and exterior conditions through constructive systems, structural modulation and material properties [35] with improved energy performance capacity, adapting modern architecture to the climate [36]. Authors grouped the buildings, and their architectonic filter/threshold, into four categories:

1. The façade intermediated by balconies: Julio Barros Barreto building (1947), Bristol building (1950) and Nova Cintra building (1948) (1,2,3 Figure 1);
2. The façade intermediated by glass planes: Nova Cintra building (1948) and Barão Gravatá building (1952) (3,4 Figure 1);
3. The façade intermediated by coupled concrete brise-soleil: MMM Roberto building (1945) and Dona Fátima and Finúsia building (1951) (5,6 Figure 1);
4. The façade intermediated by filters, cobogós, shutters and trusses: Ramirez building (1954) and Sambaíba building (1953) (7,8 Figure 1).

Our case study Nova Cintra building belongs in two categories as it uses the north façade as a filter, intermediated by balconies, and the south façade intermediated by glass planes.

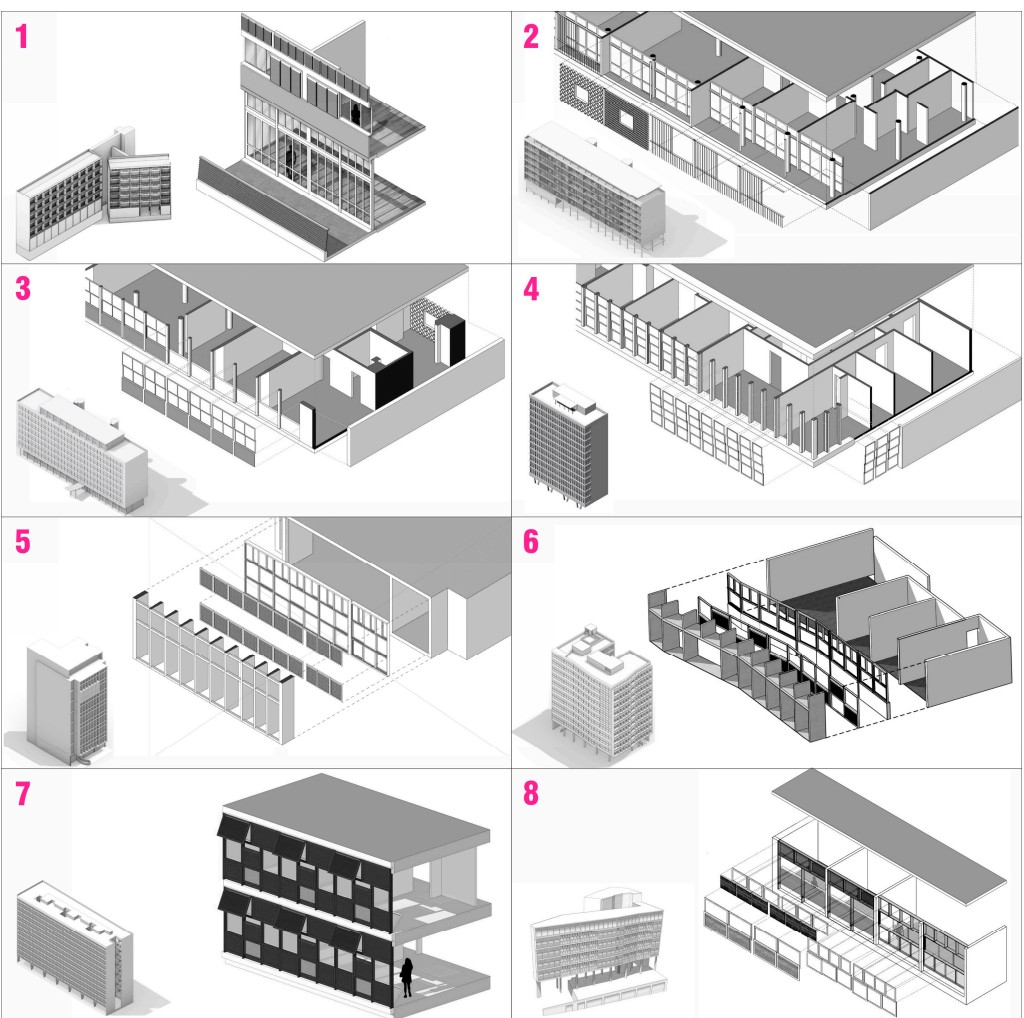

**Figure 1.** Carioca Modern Façades selected buildings, volume and façade system. **1**- Barreto building, by Irmãos Roberto, 1947; **2**- Bristol building, by Lucio Costa, 1950; **3**- Nova Cintra building, by Lucio Costa, 1948; **4**- Barão Gravatá building, by Sérgio Bernardes, 1952; **5**- MMM Roberto, by Irmãos Roberto, 1945; **6**- D. Fátima building, by Irmãos Roberto, 1951; **7**- Angel Ramirez building, by Irmãos Roberto, 1954; **8**- Sambaíba building, by Irmãos Roberto, 1953. Image adapted from Mara Eskinazi [32].

### 2.2. Case Study Description: The Nova Cintra Building

The architect Lucio Costa designed the Nova Cintra building in 1948. According to Lucas and Bastos [37], the Nova Cintra is a modern Brazilian example of passive bioclimatic adaptation. The adaptation addresses the following: (1) Climatic context—for most of the year, the city of Rio de Janeiro has a hot and humid tropical weather, with high temperature and relative humidity, with more rain in summer and dry periods in winter. This weather produces thermal discomfort and a fundamental resource to mitigate this is to have interior air renovation, combined with shading, reducing the interior thermal gain, avoiding excessive direct solar light, but taking advantage of natural daylight. (2) Building implantation—Nova Cintra main axe is longitudinal to the street ($66 \times 44$ m) with a deviation of 20 degrees to north. The building position and the façade envelope minimize the thermal load: the blind façade in the north and east; the west façade only with ventilation

areas for the toilets; (3) Wind permeability—the building orientation benefits of the region's prevailing south/southeast wind, but also of its position in the terrain, in steps, over pilotis, with an indented upper floor and especially a balcony in the north façade, with a shading system. (4) Façade and shading composition—the retreated structure should be enhanced in relation to the façade, with a 4 m metric in the north and south elevations, with internal walls aligned and perpendicular to the larger façade. The north façade shading uses brises and cobogós, followed by a balcony that separates the rooms with a sliding window frame. The balconies' shading and the internal walls' alignment favours natural ventilation, along the year, and controls excessive solar daylight, favouring passive comfort of the building, without air-conditioning systems.

The authors selected the north façade intermediated by balconies and by a shading system, mainly comprised of ceramic cobogós and concrete brise-soleil, as displayed in Figure 2.

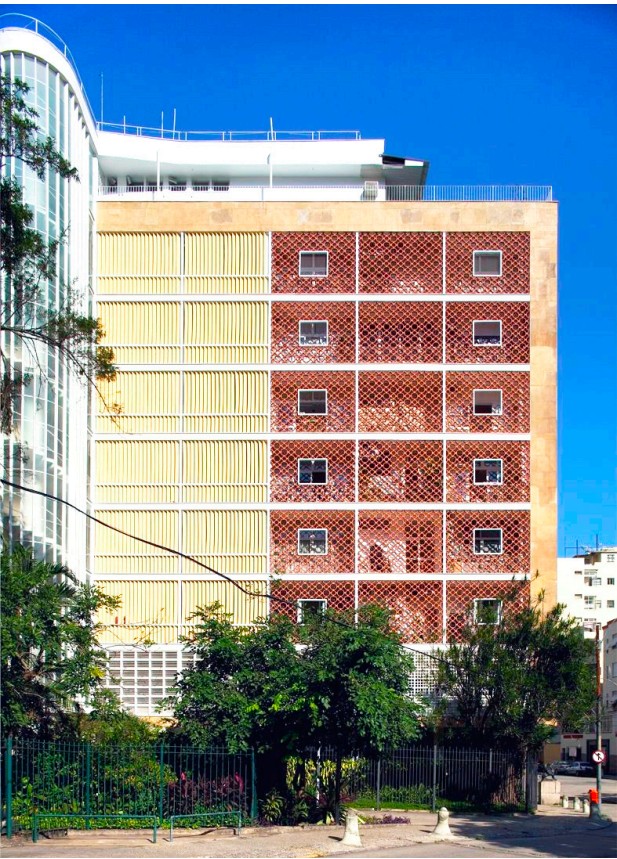

**Figure 2.** Nova Cintra building north façade, with the façade three main shading systems: the ceramic cobogó with window, the ceramic cobogó without window and the concrete brise-soleil. There are two other shading systems, the lattices and the shutters, both from wood, applied only in the building's first floor. Photo credits, the authors.

*2.3. First Research Stage: Nova Cintra 3D Modelling, North Façade Analysis, Considering Insolation and Illuminance, Combinatorial Modelling and Multi-Objective Optimization*

2.3.1. Nova Cintra 3D Modelling and Analyses

The ongoing research project Carioca Modern Façades comprises two research developments, the Nova Cintra 3D model energy performance analysis and the combinatorial modelling with multi-objective optimization, to recombine the position of the three main shading systems in the north façade. An article presented and published at the international conference SIGraDi 2022 addresses this research [32].

The research resorted to visual and textual programming using Grasshopper and Python to automate tasks. The Nova Cintra 3D model (Figure 3) adapted previous 3D models focusing on the information organization to analyse and simulate performance. The performance analyses included the following: (1) insolation (or solar irradiation), calculating the solar irradiation per unit area, in kWh/m$^2$, and the total solar irradiation, in kWh, during the year in the building envelope (roof and façades, including shading systems). Figure 4 shows the insolation in the Nova Cintra building (left) and the insolation detail in Nova Cintra's north façade (right); (2) the illuminance, after being filtered by the shading systems, in the building interior space. Using this information, we can identify the interior spaces that have appropriate daylight, between 100 and 3.000 lux [38]. Below 100 lux, there is lack of daylight and above 3.000 lux, there is excess. Higher levels of illuminance increase interior temperature and might require artificial cooling.

These analyses show building surfaces with higher and lower insolation across the year. Figure 4 shows the Nova Cintra building insolation graphical analysis. The roof surface has the higher insolation, 1.241 kWh/m$^2$, followed by the north façade, with 587 kWh/m$^2$. The roof receives the yearly higher result regarding the total value of solar irradiation (TVSI), 1.698.142 kWh (42.8% of the entire envelope), obtained by multiplying the insolation by the surface area. The roof is followed by the north façade, which obtains 972.971 kWh of TVSI (24.5% of the envelope). In the north façade, the shading system cobogó with window registers the higher insolation, 696 kWh/m$^2$, and next the cobogó without window, with 651 kWh/m$^2$, and the brise-soleil, with 495 kWh/m$^2$. However, regarding the TVSI, is the brise-soleil that achieves the higher, 339.656 kWh (75.2% total shading), due to a higher surface area, 682 m$^2$, followed by cobogó with window, with 68.924 kWh (15.3%) and 101 m$^2$, and the cobogó without window, with 43.057 kWh (9.5%) and 66 m$^2$.

The research then analysed the interior rooms' performances from the first, fourth and seventh (and last) floor, near the north façade, regarding illuminance. To calculate the cases for the other floors, the research resorted in interpolation, to diminish the analysis time. Analyses comprehend winter solstice (21st June at 12 p.m.), summer solstice (21st December at 12 p.m.) and autumn equinox (20th March at 12 p.m.), with the Sun in intermediate position. For the winter solstice, the average illuminance is 4.152 lux. The value decreases for 1.528 lux in the autumn equinox, reaching 640 lux in summer solstice. As such, in the summer solstice and in the autumn equinox, the illuminance is adequate, between 100 and 3.000 lux. For the winter solstice, the average exceeds 3.000 lux, which can provoke overheating and require artificial cooling. Thus, despite the shading systems not being efficient throughout the whole year, they have a sun-blocking capacity. Figure 5 displays the illuminance for the intermediate floor (the 4th), with and without shading (image below at left and right).

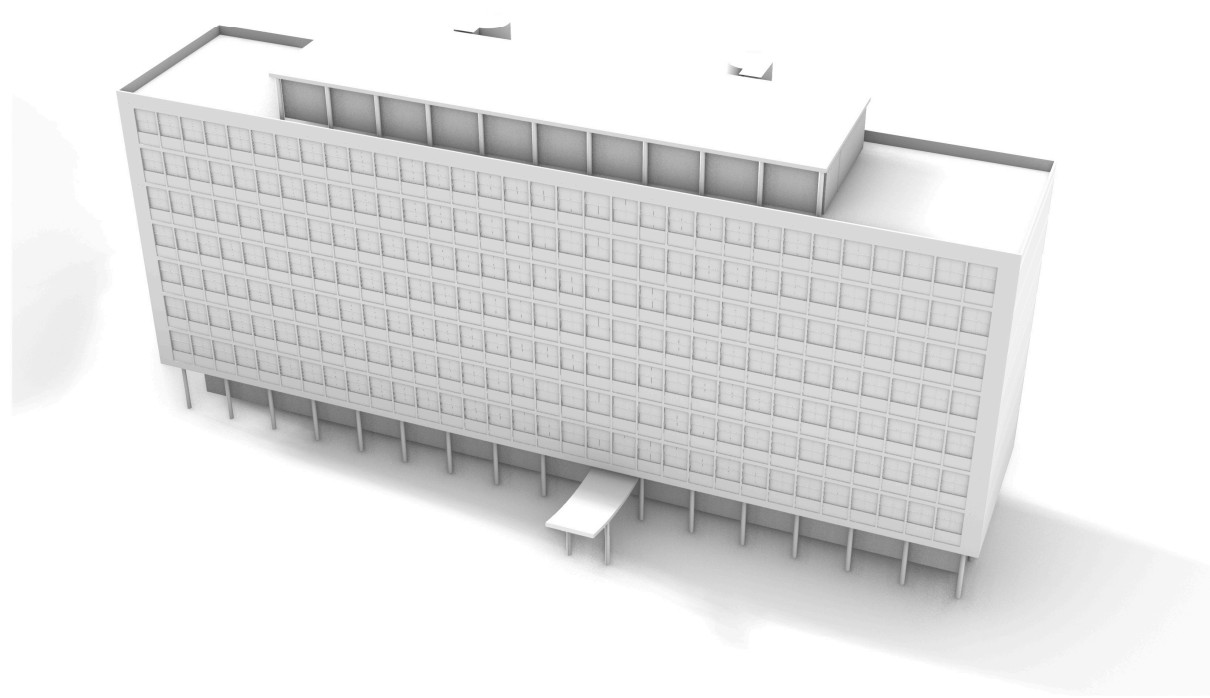

**Figure 3.** 3D model of Nova Cintra building.

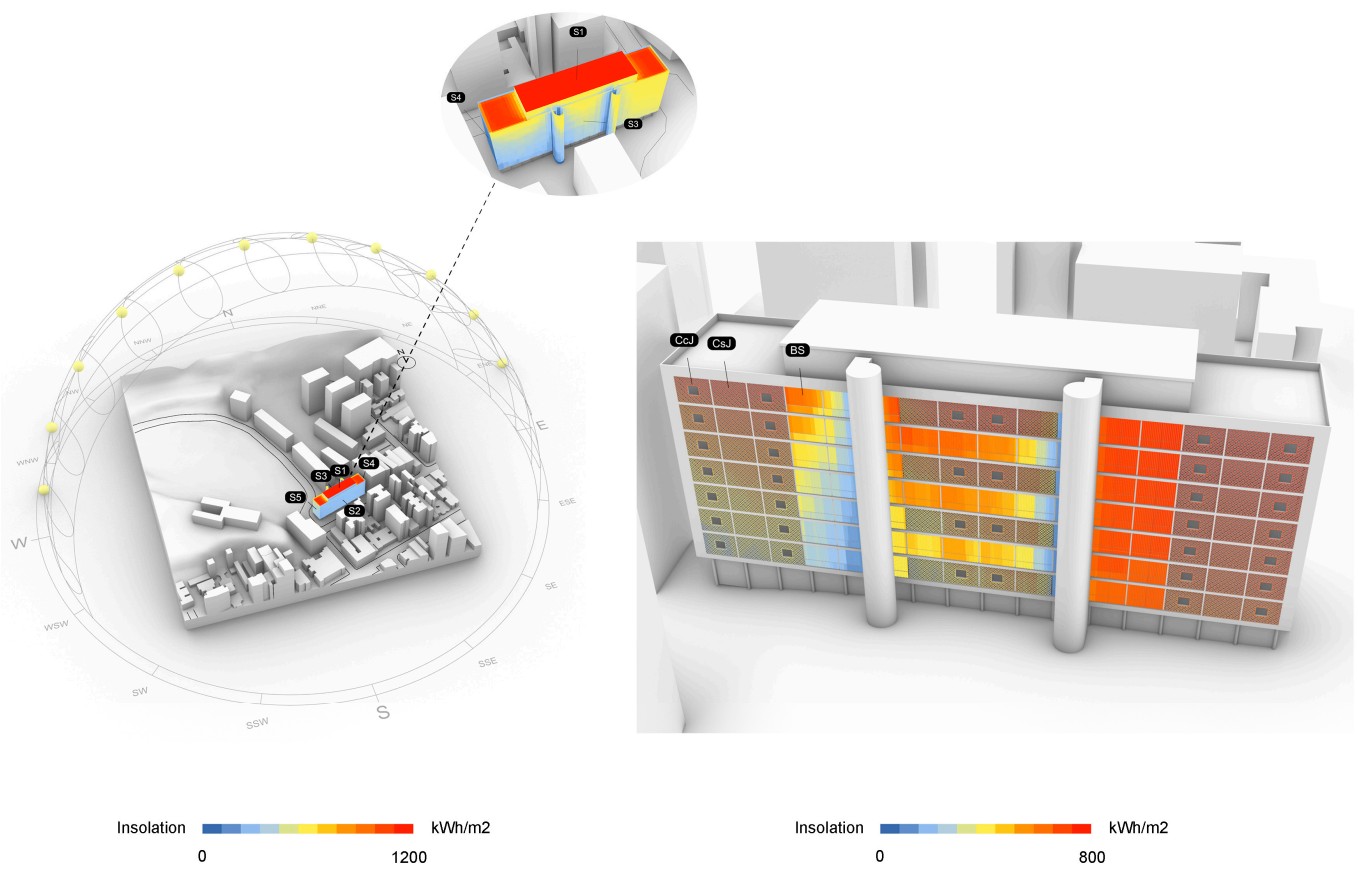

**Figure 4.** Nova Cintra building envelope insolation (**left**) with a north façade detailed view (**above**) and insolation in the shading systems of the north façade (**right**).

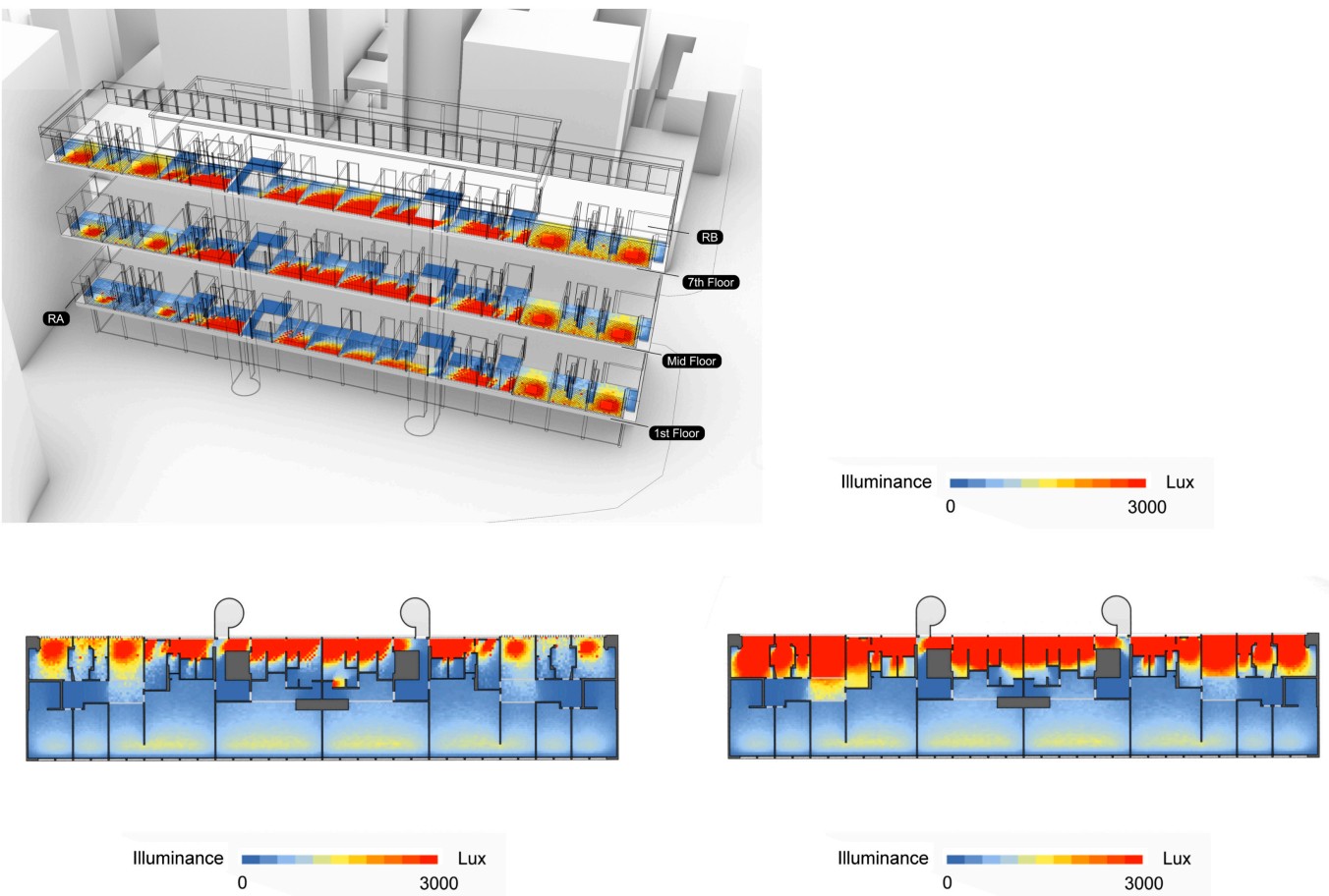

**Figure 5.** Nova Cintra building: illuminance in winter solstice (21st June) for the rooms of the 1st, 4th and 7th floors, on the north façade (**above left**); illuminance in the 4th floor in winter solstice (21st June), with and without the existing shading systems (**below left** and **right**).

### 2.3.2. Detailed Analyses, Combinatorial Modelling and Multi-Objective Optimization

After analysing the Nova Cintra building envelope for insolation and illuminance, the research performed detailed analyses of the north façade's shading systems for insolation and illuminance. For insolation and illuminance, analyses rely on the Ladybug and Honeybee plug-ins (versions 1.2.0 and 0.0.63, respectively). The initial insolation analyses comprised the whole year, while illuminance analyses considered the winter solstice (21st June at 12 p.m.), summer solstice (21st December at 12 p.m.) and autumn equinox (20th March at 12 p.m.), to decrease calculation requirements. The solstice's results are the best- and worst-case scenarios, regarding illuminance performance.

The detailed analyses evaluated the north façade's performance, composed of three main shading types: cobogó with window (identified with Letter A), cobogó without window (Letter B) and brise-soleil (Letter C). This discarded two shading systems, wood lattice and wood shutter, were used only on the first floor, for commercial use. The three shading systems's detailed analyses address the winter and summer solstices, regarding insolation and illuminance, at 10 a.m., 12 a.m. and 4 p.m.

To calculate individual performance, we performed individual façade analysis for each shading system (letters A, B and C) comprising the $14 \times 7 = 98$ north façade modules. Calculating for each shading system: (1) insolation in the exterior face of the shading elements, in $Wh/m^2$; (2) insolation in the plane immediately after the shading elements, in $Wh/m^2$; (3) insolation in the plane after the shading elements, but without shading, in $Wh/m^2$; (4) average illuminance in the north façade rooms, in lux; and (5) points number in the north façade rooms, with illuminance between 300 and 750 lux.

Detailed analyses feed a combinatorial modelling process to select the best shading systems, in each module, to improve the building performance by minimizing solar radiation in interior rooms (objective 1), maximizing the points number with adequate illuminance (objective 2) and maximizing average illuminance (objective 3), to assure adequate interior daylight. To automate data analyses with visual and textual programming in Python, we created an alphanumeric label to identify each simulation, such as "CintraWkwmCIE21jun10:00_LetraA" with the reference to the building name, type of performed analysis, analysis day and hour. Concerning the three objectives mentioned, we realized $1 \times 2 \times 1 \times 2 \times 3 \times 3 = 36$ simulations considering the building envelope in the urban context, climatic data and materials used in the construction. To interpret the emergent façade patterns, we developed a graphic representation, using colour gradients in meshes.

To compare the results, the research normalized results on a scale of 0 to 1. The research tested different methods to find the best performance for each scenario (21st June and 21st December, at 10 a.m., 12 p.m. and 4 p.m.) for the 98 modules of the north façade. Initially, we tested random combinations of the shading types letters (A, B, and C) using genetic algorithms to maximize the performance, through the use of the multi-objective optimization plug-in Octopus in Grasshopper. After, a quick sorting algorithm with conditionals proved more straightforward in selecting the best performance for each module. The façade solutions correspond to a list of 98 letters A, B and C. The textual algorithm in Python (inside grasshopper), compares the results using different weights for each objective, in a sensitivity analysis, to understand how each factor affects global selection and the results range.

The multi-objective optimization results are present in Section 3 Results.

*2.4. Second Research Stage: Parametric and Algorithmic–parametric Modelling, Single-Objective and Multi-Objective Optimization*

The Carioca Modern Façades second research stage model parametrically and algorithmically parametrically the Nova Cintra north façade shading systems. A single-objective and a multi-objective optimization, regarding insolation, illuminance and air temperature, looks for formal configurations that improve performance. Finally, we compare the performance of the modelled shading systems obtained through the combinatorial, parametric and algorithmic–parametric modelling processes used together with the optimization methods. The purpose is to identify the advantages of each modelling process.

2.4.1. Parametric Modelling and Performance Optimization

The parametric modelling process transforms an existing shading system, the cobogó with window, labelled letter A. The change affects two parameters: the central window amplitude and the circular components' thickness amplitude of the original cobogó. This simple parametric definition enables quick insolation and illuminance optimization. The optimization considers, together with insolation and illuminance, a third energy parameter: interior air temperature.

The new parametric cobogó intends on avoiding overheating in the interior compartment. Figure 6 displays cobogó with window type A parameters.

The performance of this cobogó in the north façade upper left corner module, considers: (1) insolation in the cobogó superior face, in kWh/m$^2$ and kWh; (2) number of room points with illuminance between 300 and 750 lux; and (3) room temperature in °C, using Honeybee plug-in (version 1.6.0).

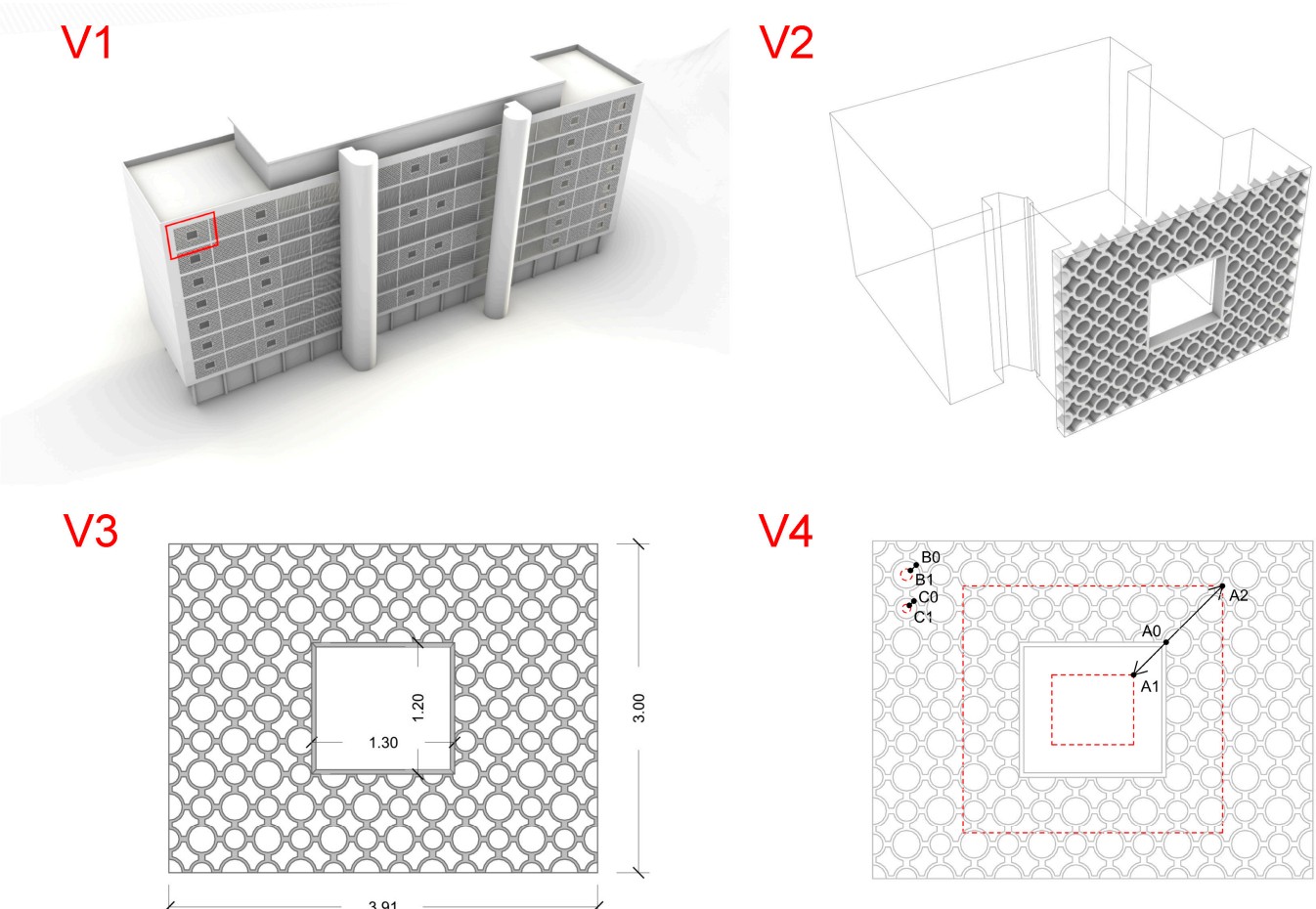

**Figure 6.** Shading system cobogó with window (type A) located upper left corner module (Nova Cintra north façade) (image **V1**); axonometric perspective cobogó with window (image **V2**); front view cobogó with window (image **V3**); parameterization cobogó with window. Parameter 1: central window amplitude, smaller (point A0 to A1), or larger (point A0 to A2), ranging from 0.0 to 1.8; Parameter 2: thickness amplitude of largest and smallest circles of the cobogó (points B0 to B1 and C0 to C1, respectively), ranging from 0.0 to 1.0 (image **V4**).

Single-objective optimization uses separated performance analyses for the cobogó type A, regarding insolation, illuminance and air temperature. After single-objective optimization, multi-objective optimization was conducted to improve all goals simultaneously. Single-objective optimization relies in Galapagos (version 0.2.0448) and multi-objective in Octopus (version 0.4), both plug-ins for Grasshopper. Galapagos [39] is a plug-in that uses genetic algorithms to discover the optimal combination of values for a given set of variables, applying the Darwinian theory of evolution on the design of alternatives. The result after several iterations and the elimination of unfit solutions is a pool of optimized design alternatives [17] for an objective, a single function, hence the use of Galapagos for single-objective optimization. Octopus [40] is a plug-in that utilizes a genetic algorithm for multi-objective (multiple function) optimization, named the SPEA-2 algorithm, published by Zitzler, Laumanns and Thiele [41]. There is a description of the implementation of the algorithm in the plug-in in the work of Vierlinger [42]. Galapagos and Octopus are therefore used in this research to obtain optimized results for the energy performance of the developed shading system, cobogó type A, for insolation, illuminance and air temperature. Section 3 presents the Cobogó type A results using single and multi-objective optimizations.

2.4.2. Algorithmic–Parametric Modelling and Performance Optimization

To improve cobogó type A energy performance, the original shape underwent additional transformations, using the Building Envelopes Grammar [30,31]. The Building Envelopes Grammar is an algorithmic–parametric modelling process. It contains different shape creation and transformation operations, which form the algorithmic part of the modelling process. Combinations can follow different arrangements, using different algorithms to form efficient buildings to harvest solar energy.

The cobogó type A transformation uses the normalization and division operations of the Building Envelopes Grammar. Transformations constitute the algorithmic part of the algorithmic–parametric modelling process. The normalization operation transforms a target surface into a surface perpendicular to the sun's rays, in a specific geographic location and time. Surfaces perpendicular to the sun's rays are more efficient at harvesting solar energy. The division operation divides the surface into two equal surfaces.

Next, it is necessary to divide a created plane, coincident with the exterior face of the cobogó type A, present in the upper left corner module (Nova Cintra north façade), in two equal horizontal surfaces. The created plane ranges from the floor to the ceiling and from the left room wall to the right room wall. The top horizontal surface requires normalization, but to the bottom horizontal surface does not, to allow the solar rays to reach the floors below. The top horizontal surface normalization considers the Sun's positions across the year, hourly, located more to north than the north façade. Another constraint considered for the normalized surfaces was that their four vertices should not be more than 1.5 m away from the façade's exterior plane.

Later, we adjusted the obtained set of possible normalized surfaces, moving each surface from their point located at a greater distance inside the building, to their perpendicular point, coincident with the exterior façade plane. This process assures that the selected set of normalized surfaces are totally outside the building, with only one vertex coincident with the north façade plane. Next, we moved the upper vertices of the bottom horizontal divided surface to make them coincident with the lower vertices of the normalized surfaces. This procedure assures a set of bottom horizontal surfaces, each one connected to a normalized surface. Figure 7 presents a schematic explanation of the division and normalization operations, applied on the upper left corner module.

Research then applies the parametric part of the process selecting, from the set of normalized surfaces, the normalized surface with the best performance for solar energy harvesting (insolation). Best performance relies on single-objective optimization using Galapagos.

After selecting the normalized surface with the best insolation performance, the algorithmic procedure returns, being the original cobogó type A projected in the selected normalized surface and the bottom horizontal surface associated. Current research developed the projection operation of cobogó type A, that is now part as well of the Building Envelopes Grammar. The algorithmic procedure of the algorithmic–parametric modelling finishes with the conclusion of the projection operation.

After finished the algorithmic part, the parametric part of the algorithmic–parametric process is also concluded, being realized with additional single-objective optimizations using Galapagos, considering parametric variations for the projected cobogó. The parameters are those adopted in the previous parametric modelling process (Section 2.4.1): the central window amplitude and the circular components' thickness amplitude. The optimization considers insolation, illuminance and temperature, separately.

Octopus performs additional multi-objective optimization for the normalized and parameterized cobogó, considering simultaneously insolation, illuminance and temperature. The next section presents the single- and multi-objective optimizations results.

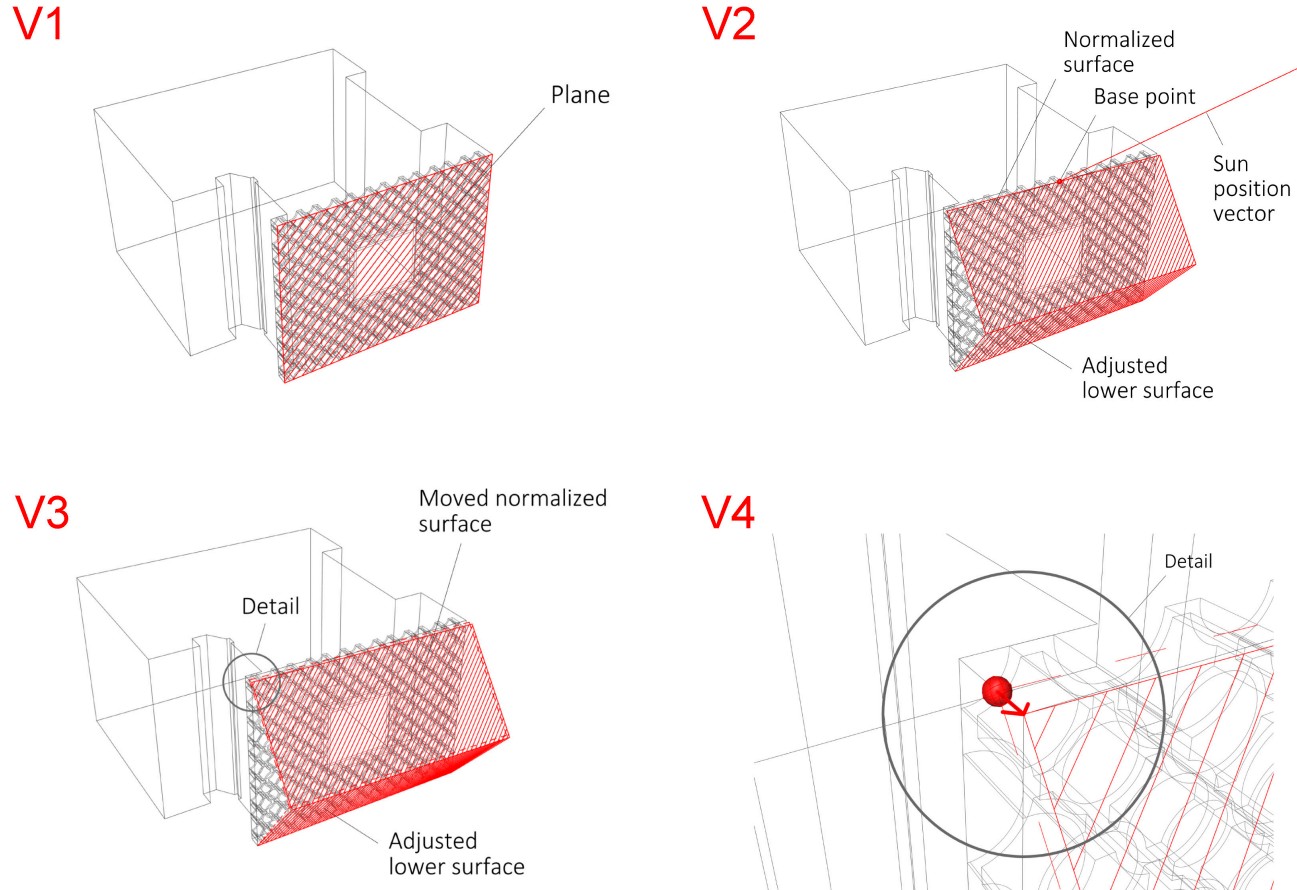

**Figure 7.** Plane coincident with exterior cobogó face (image **V1**). Plane division in two equal horizontal surfaces, top horizontal surface normalization, turning it perpendicular to one sun position (in this example, May 2nd at 12 p.m.), considering as base point for normalization operation the midpoint of the plane upper edge; adjustment of the bottom surface, connecting their two upper vertices to the normalized surface two lower vertices (image **V2**). Movement of normalized surface, from the point located more in the interior of the room to the perpendicular point located in the plane coincident with the cobogó exterior face (image **V3**). Detailed view of the movement of normalized top surface, showing the movement reference points (image **V4**).

## 3. Results

This section presents the results of the first and second stages of the Carioca Modern Façades research:

1.  The results achieved in combinatorial modelling, recombining the original shading systems in the Nova Cintra north façade, used together with multi-objective optimization, considering insolation and illuminance measures. The results achieved in insolation and illuminance are also highlighted for the best performance shading system, obtained through the combinatorial modelling process, in the upper left corner module (Nova Cintra north façade). We use this module in the parametric and algorithmic–parametric modelling and optimization processes. These last results will enable us to compare the results obtained for this module in all the modelling and optimization processes used in the first and second stages of the research.
2.  The original shading type results of the upper left corner module (Nova Cintra north façade) considering insolation, illuminance and air temperature. These results enable a comparison of the original shading performance with the performance of the shading developed with the modelling and optimization processes.

3. Results using parametric and algorithmic–parametric modelling processes over the original shading system cobogó with window (type A), together with single-objective and multi-objective optimization, regarding insolation, illuminance and air temperature.

### 3.1. Results of the Combinatorial Modelling and Multi-Objective Optimization

The Nova Cintra north façade results, using combinatorial modelling together with multi-objective optimization, are relative to three shading types: the cobogó with window (letter A), cobogó without window (letter B) and brise-soleil (letter C), which are displayed in Table 1. The multi-objective optimization considers three goals: (1) to minimize insolation after the shading, in Wh/m$^2$; (2) maximize points number with ideal illuminance between 300 and 750 lux in the interior rooms; and (3) maximize the interior rooms' average illuminance, in lux. Results without ponderation (the three first columns of Table 1: A, B, C) that consider equally the scores for insolation, points number with ideal illuminance and average illuminance (ins 1, illuPts 1, illuAvg 1), show that both in June (Jun) and December (Dec), the letter B prevails, at 10 a.m. and 12 p.m., while at 4pm (Jun and Dec), the best solution is letter C.

**Table 1.** Multi-objective analysis of the shading types (A, B, C) for insolation, points number with ideal illuminance between 300–750 lux and average illuminance.

| Letters | A | B | C | A | B | C | A | B | C |
|---|---|---|---|---|---|---|---|---|---|
| MULTI-OBJECTIVE | 1 ins | 1 illuPts | 1 illuAvg | 0.2 ins | 0.6 illuPts | 0.2 illuAvg | 0.2 ins | 0.2 illuPts | 0.6 illuAvg |
| Jun 10 a.m. | 0 | 88 | 10 | 6 | 72 | 20 | 4 | 53 | 41 |
| Jun 12 p.m. | 0 | 97 | 1 | 0 | 97 | 1 | 0 | 12 | 86 |
| Jun 4 p.m. | 1 | 30 | 67 | 12 | 30 | 56 | 1 | 30 | 67 |
| Dec 10 a.m. | 0 | 95 | 3 | 0 | 95 | 3 | 0 | 95 | 3 |
| Dec 12 p.m. | 29 | 47 | 22 | 36 | 25 | 37 | 29 | 47 | 22 |
| Dec 4 p.m. | 36 | 20 | 42 | 58 | 2 | 38 | 36 | 20 | 42 |
| SUM | 66 | 377 | 145 | 112 | 321 | 155 | 68 | 257 | 257 |
| AVERAGE | 1 | 66 | 16 | 9 | 51 | 29 | 3 | 39 | 42 |

Taking more in consideration the objective points with ideal illuminance (ins 0.2, illuPts 0.6, illuAvg 0.2), solution B continues to prevail, with the exceptions of 4 p.m. Jun and 12 p.m. Dec where the best solution is C, and of 4 p.m. Dec, where letter A prevails.

Finally, favouring the objective average illuminance (ins 0.2, illuPts 0.2, illuAvg 0.6), there are three best solutions with the letter C (Jun 12 p.m. and 4 p.m., Dec 4 p.m.), with letter B being better in the other three situations (Jun 10 a.m., Dec 10 a.m. and 12 p.m.).

In summary, letter B, cobogó without window, is the best global solution. Letter C is the second best solution, and there are less situations in which the letter A dominates. We present a graphical representation of Table 1, using the corresponding shading types, in Figure 8. Through this graphical representation, it is possible to verify that in Dec 4 p.m., considering the different ponderations, letter A (red) is the best solution for many modules, suggesting low illuminance in this period, due to the Sun's high position. In the afternoon periods letter C (blue), brise-soleil, is also frequent, especially in Jun 4 p.m.

In the upper left corner module (Nova Cintra north façade), used in the other modelling and optimization processes of the research, letter B (cobogó without window, treated by type B in the other modelling and optimization processes) is the one that achieves the best results for the June 12 p.m. and December 12 p.m. periods. These are the periods analysed in the other modelling and optimization processes of the research.

Table 2 displays insolation, illuminance and also air temperature obtained for the shading cobogó without window (type B), in the upper left corner module (Nova Cintra north façade), for the periods of June 12 p.m. and December 12 p.m.

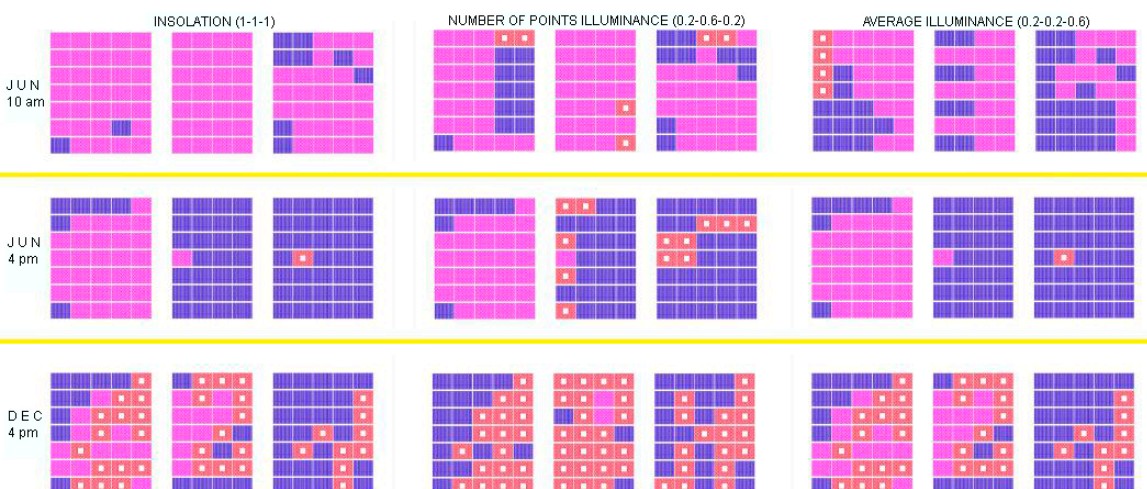

**Figure 8.** Graphical results combinatorial modelling and multi-objective optimization of the shading types letter A (red), B (lilac) and C (blue), resulting in a string (ACCBAB...) optimized according to objectives and weights.

**Table 2.** Insolation, points number with ideal illuminance and air temperature, simulated for the shading cobogó without window (type B), located at upper left corner module (Nova Cintra north façade).

| Shading Analyses: Cobogó without Window (Type B) | Insolation (kWh ǀ kWh/m²) | Nº of Points with Ideal Illuminance (300–750 lux) ǀ Total Number of Points | Air Temperature (°C) |
|---|---|---|---|
| Winter solstice, 21st June 12 p.m. | 1.89 kWh ǀ 0.63 kWh/m² | 23 ǀ 178 points | 28.35 °C |
| Summer solstice, 21st December 12 p.m. | 0.44 kWh ǀ 0.15 kWh/m² | 64 ǀ 178 points | 33.62 °C |

Shading type B presents the following results: in June 12 p.m., 1.89 kWh and 0.63 kWh/m² for insolation, 23 points with illuminance between 300 and 750 lux in a total of 178 possible points and 28.35 °C of air temperature; in December 12 p.m., 0.44 kWh and 0.15 kWh/m² for insolation, 64 points with illuminance between 300 and 750 lux in a total of 178 possible points and 33.62 °C of air temperature.

Research discards average illuminance here, as in the other cases addressed.

Regarding insolation, the values considered for the combinatorial modelling and multi-objective optimization consider a plane behind the interior face of the shading. However, the insolation indicated in Table 2 is relative to that obtained in the exterior face of the shading, meaning that there is referent to sun energy harvesting, that can be used for the generation of renewable energy through the use of photovoltaic technology. The research considers this in the other modelling and optimization processes, hence their indication in Table 2, in order to be comparable with the insolation obtained in the other modelling and optimization processes of the research.

The insolation in the exterior face of the shading relates directly with the insolation behind the interior face of the shading: the bigger the insolation in the shading exterior face, the smaller the corresponding insolation behind the shading.

*3.2. Results Original Shading Type, Cobogó with Window*

We selected one module, the upper left corner module of the Nova Cintra building north façade, and their original shading type A (cobogó with window), for the realization of the other modelling and optimization processes of the research, due to the following reasons:

1.  To save time in the development of the single-objective and multi-objective optimizations for insolation, illuminance and air temperature, for the parametric and algorithmic–parametric modelling processes;
2.  The selected shading type A, cobogó with window, is the original of the upper left corner module of the Nova Cintra north façade and is the shading type that has the worst results in the combinatorial modelling and multi-objective optimization; so, it needs improvement;
3.  The upper left corner module is the first module in the façade, if we interpret the façade as a mathematical matrix or as a sentence that starts from top to bottom, from left to right.

Therefore, we obtained the results for the original shading type A, regarding insolation, illuminance and air temperature, in the upper left corner module of the Nova Cintra north façade, in order to compare them with the results obtained for the combinatorial, parametric and algorithmic–parametric modelling processes. Table 3 shows results obtained for the original shading type A, cobogó with window: 2.26 kWh for insolation and 0.63 kWh/m$^2$ for insolation efficiency indicator, regarding shading harvesting by square meter, in the exterior face of the shading, 28 points with ideal illuminance (between 300 and 750 lux) out of 178 possible points and 29.20 °C of air temperature, during the winter solstice, 21st June 12 p.m.

**Table 3.** Results of cobogó with window (type A), for the winter and summer solstices: insolation, points number with ideal illuminance and air temperature.

| Shading Analyses: Original Cobogó with Window (Type A) | Insolation (kWh ∣ kWh/m$^2$) | N° of Points with Ideal Illuminance (300–750 lux) ∣ Total Number of Points | Air Temperature (°C) |
|---|---|---|---|
| Winter solstice, 21st June 12 p.m. | 2.26 kWh ∣ 0.63 kWh/m$^2$ | 28 ∣ 178 points | 29.20 °C |
| Summer solstice, 21st December 12 p.m. | 0.52 kWh ∣ 0.10 kWh/m$^2$ | 112 ∣ 178 points | 34.51 °C |

In the summer solstice, 21st December 12pm, the results are 0.52 kWh and 0.10 kWh/m$^2$ for insolation, 112 points with ideal illuminance (between 300 and 750 lux) out of 178 possible points and 34.51 °C air temperature.

Figure 9 displays a graphical representation of the results of insolation, illuminance and air temperature, for the shading type cobogó with window, at the upper left corner module (Nova Cintra north façade), for the winter and summer solstices.

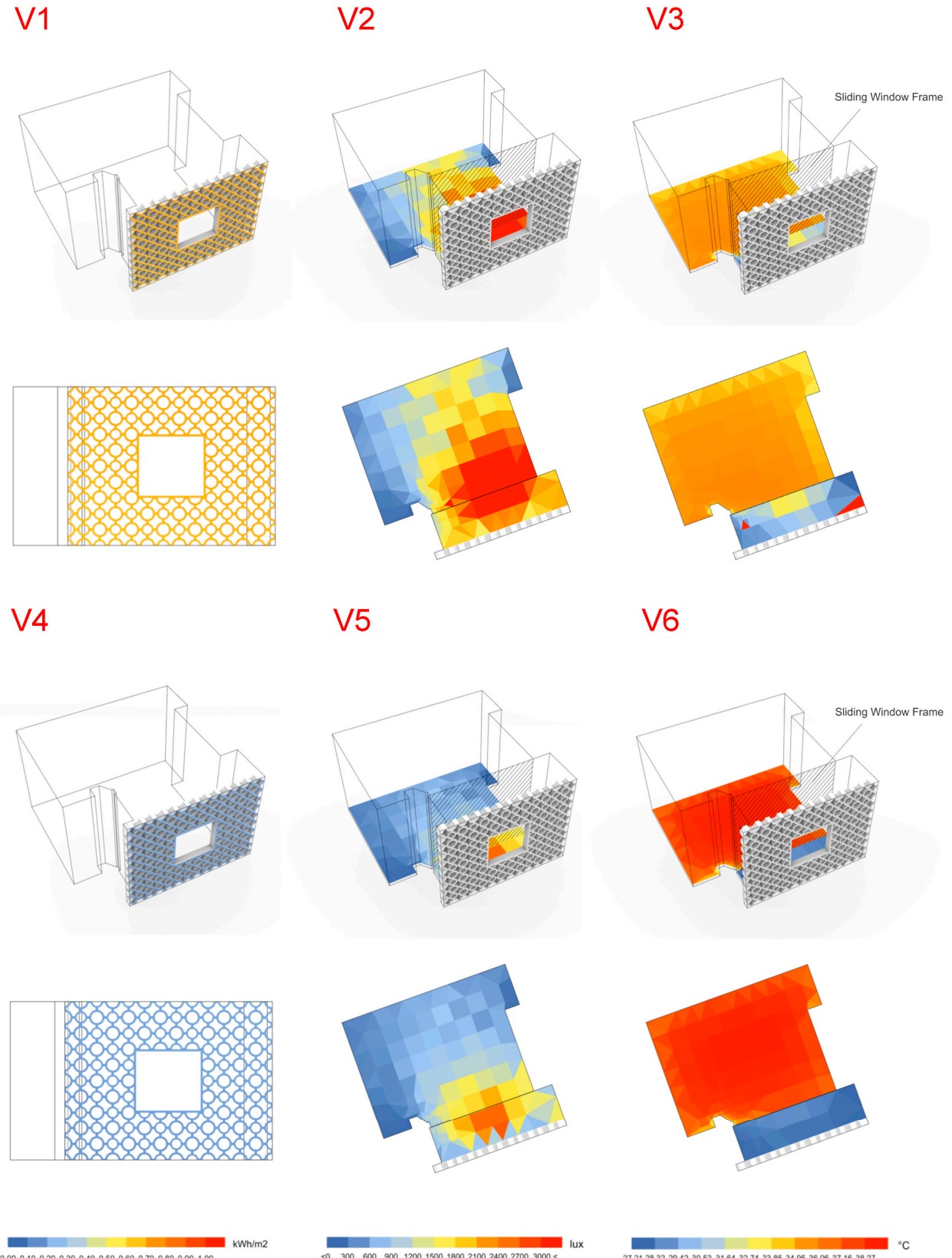

**Figure 9.** Results insolation (left column), points number with ideal illuminance (middle column) and air temperature (right column) original shading cobogó with window (type A), winter solstice, June 21st 12 p.m. (images **V1**, **V2** and **V3**) and summer solstice, December 21st 12 p.m. (images **V4**, **V5** and **V6**).

### 3.3. Results of the Parametric Modelling, with Single-Objective and Multi-Objective Optimization

The parametric modelling, as defined in Section 2.4.1, uses two parameters for the single-objective optimization of the shading type cobogó with window (type A).

The universe of solutions is defined by parameter 1, the window central void amplitude, with values that range from 0.0 to 1.8, with the value 1.0 being the original size of the cobogó with window (1.30 m length and 1.20 m height); and parameter 2, the circular modules' thickness amplitude, that range from 0.0, the maximum thickness (0.145 m in the bigger circles and 0.118 m in the smaller circles) to 1.0, the minimum and original thickness (0.04 m in the bigger circles and 0.035 m in the smaller circles).

Table 4 presents the single-objective optimization results obtained for insolation, points number with ideal illuminance (300–750 lux) and air temperature, for the winter and summer solstices. For instance, the single-objective optimization of the insolation for the winter solstice has a result of 4.77 kWh and 0.62 kWh/m$^2$, with 0.1 (0.13 m length and 0.12 m height) for parameter 1 and 0.0 (thickness of 0.145 m in bigger circles and of 0.118 m in smaller circles) for parameter 2. The single-objective optimization of the insolation for the summer solstice presents the result of 1.11 kWh and 0.14 kWh/m$^2$, with 0.0 (0 m length per 0 m height) for parameter 1 and 0.0 (thickness of 0.145 m in bigger circles and of 0.118 m in smaller circles) for parameter 2.

Figure 10 displays a graphical representation with colours of the results of the single-objective optimization processes, for the winter and summer solstices.

**Table 4.** Single-objective optimization results of the parametric model realized for the cobogó with window (type A), regarding insolation, points number with ideal illuminance and air temperature, for the winter and summer solstices.

| Single-Objective Optimization of the Parametric Model | Parameter 1: Central Window Ampitude | Parameter 2: Circular Thickness Amplitude | Insolation (kWh \| kWh/m²) | Points number with Ideal Illuminance (300–750 lux) \| Total Number of Points | Air Temperature (°C) |
|---|---|---|---|---|---|
| Winter solstice, 21st June 12 p.m. | 0.1 | 0.0 | 4.77 kWh \| 0.62 kWh/m$^2$ | - | - |
| | 0.1 | 0.3 | - | 95 \| 178 points | - |
| | 0.5 | 0.0 | - | - | 28.34 °C |
| Summer solstice, 21st December 12 p.m. | 0.0 | 0.0 | 1.11 kWh \| 0.14 kWh/m$^2$ | - | - |
| | 0.3 | 1.0 | - | 118 \| 178 points | - |
| | 0.5 | 0.0 | - | - | 33.62 °C |

The research performed as well a multi-objective optimization for the shading cobogó with window (type A) to achieve an optimized shape simultaneously considering insolation, points number with ideal illuminance and air temperature performance.

Figure 11 shows a perspective of the multi-objective optimized shape obtained for the shading cobogó with window, illustrating with colour the insolation-optimized result for the winter solstice.

Table 5 shows the results obtained in the multi-objective optimization realized for the parametric model of the cobogó with window (type A), regarding insolation, points number with ideal illuminance and air temperature, in the winter and summer solstices. The winter solstice results are 4.67 kWh and 0.63 kWh/m$^2$ for insolation, 106 out of 178 possible points for ideal illuminance, 29.09 °C for air temperature, 0.1 (0.13 m length and 0.12 m height) for parameter 1 and 0.2 (thickness of 0.124 m for bigger circles and of 0.101 m for smaller circles) for parameter 2. Summer solstice results are 1.03 kWh and 0.10 kWh/m$^2$ for insolation, 96 out of 178 possible points for ideal illuminance, 33.63 °C for air temperature, 0.8 (1.04 m length and 0.96 m height) for parameter 1 and 0.0 (thickness of 0.145 m in bigger circles and of 0.118 m in smaller circles) for parameter 2.

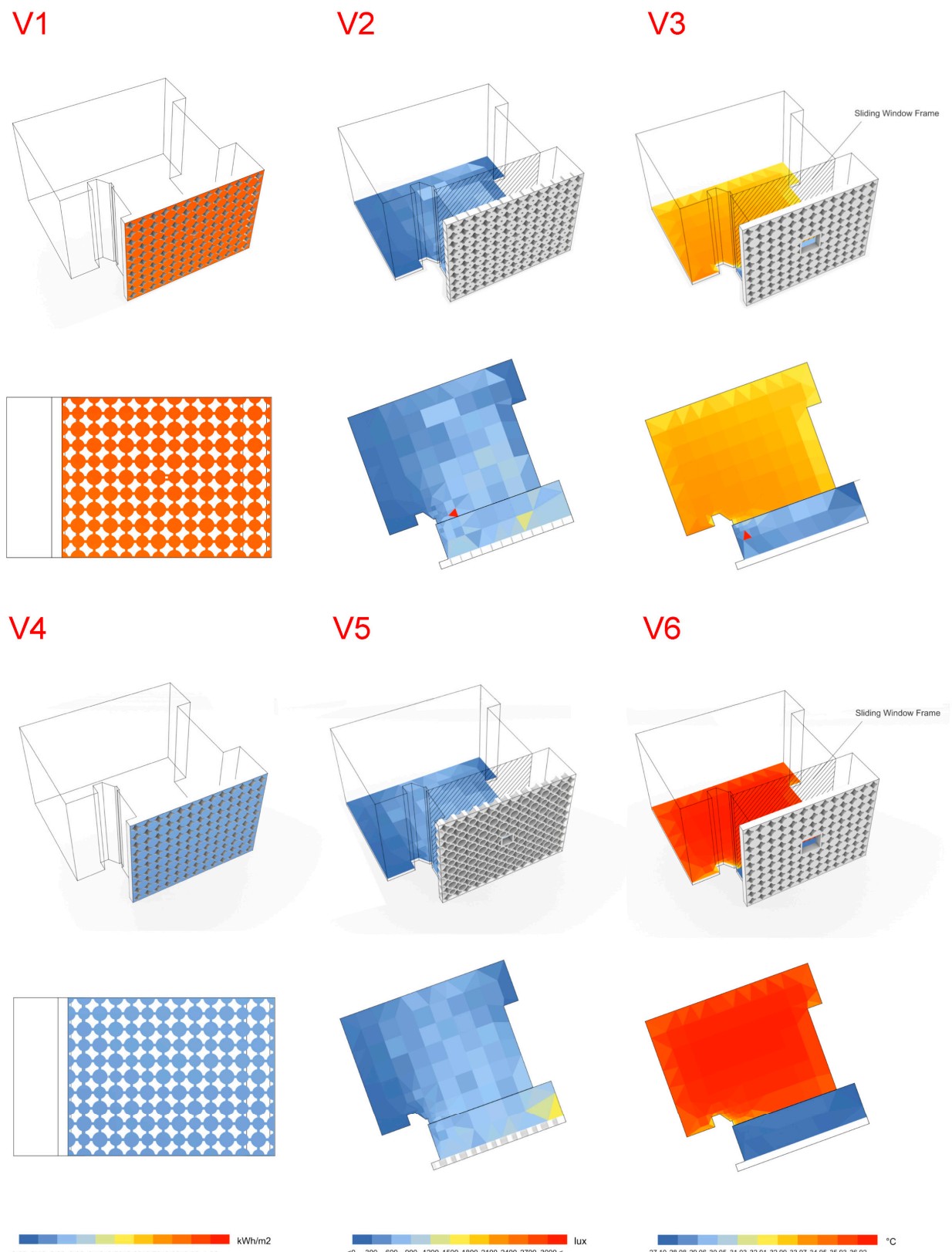

**Figure 10.** Results of the single-objective optimizations for winter solstice, June 21st 12 p.m. (images **V1**, **V2** and **V3**) and summer solstice, December 21st 12 p.m. (images **V4**, **V5** and **V6**). Results regarding insolation (**left** column), points number with ideal illuminance (**middle** column) and air temperature (**right** column).

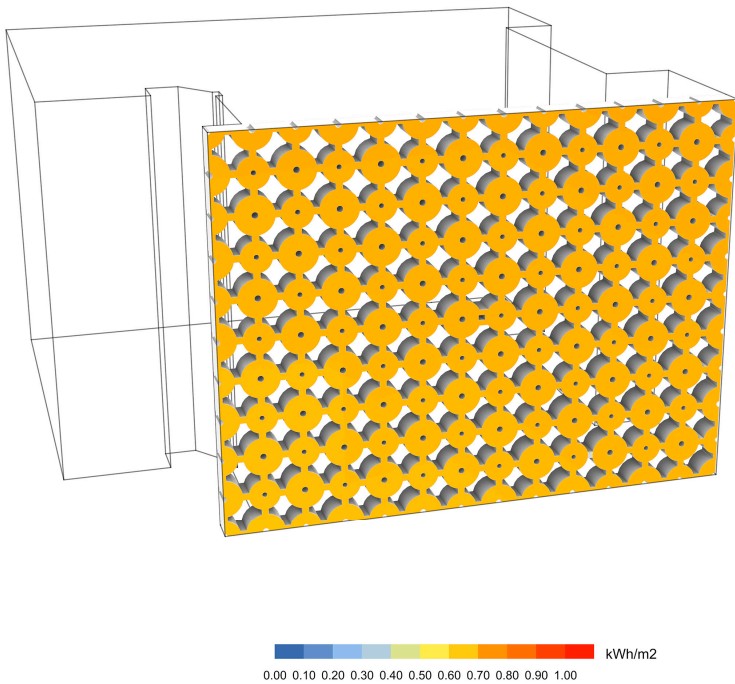

kWh/m2

0.00 0.10 0.20 0.30 0.40 0.50 0.60 0.70 0.80 0.90 1.00

**Figure 11.** Multi-objective optimization result of the parametric model realized for the cobogó with window (type A), regarding insolation, points number with ideal illuminance and air temperature, in the winter solstice, 21st June 12 p.m.

**Table 5.** Results of the multi-objective optimization realized for the parametric model of the cobogó with window, regarding insolation, points number with ideal illuminance and air temperature.

| Multi-Objective Optimization of the Parametric Model | Parameter 1: Central Window Amplitude | Parameter 2: Circular Thickness Amplitude | Insolation (kWh \| kWh/m²) | Points Number with Ideal Illuminance (300–750 lux) \| Total Number of Points | Air Temperature (°C) |
|---|---|---|---|---|---|
| Winter solstice, 21st June 12 p.m. | 0.1 | 0.2 | 4.67 kWh \| 0.63 kWh/m² | 106 \| 178 points | 29.09 °C |
| Summer solstice, 21st December 12 p.m. | 0.8 | 0.0 | 1.03 kWh \| 0.10 kWh/m² | 96 \| 178 points | 33.63 °C |

*3.4. Results of the Algorithmic–Parametric Modelling with Single-Objective and Multi-Objective Optimization*

The algorithmic–parametric modelling processes of division and normalization, described in Section 2.4.2, generate 350 possible positions to normalize the top horizontal surface, considering a planar surface from the upper left corner module (Nova Cintra north façade), coincident with the exterior surface of the cobogó with window (type A) and divided in two equal horizontal surfaces, top and bottom surfaces. The 350 possible positions for normalize the top horizontal surface, aligning this surface perpendicular to different sun positions across the year, are optimized regarding insolation, to maximize solar energy harvesting. We performed a single-objective optimization process for the winter and summer solstices. The Sun normalization's best performance is on September 1st at 12 p.m., for both solstices. Table 6 displays the best insolation performance results for a normalized top horizontal surface, obtained through the single-objective optimizations. After the selection of the normalized top horizontal surface with best performance and their associated bottom horizontal surface, the parametric model realized for the original cobogó with window (type A) is projected in these two surfaces. With this projection the algorithmic part of the algorithmic–parametric modelling process is complete.

**Table 6.** Best insolation performance for winter and summer solstices, using single-objective optimization for the normalized top horizontal surface, considering a divided planar surface from the upper left corner module (Nova Cintra north façade).

| Normalization Cobogó with Window (Type A) | Sun Position | Insolation (kWh/m² ǀ kWh) |
|---|---|---|
| Winter solstice, 21st June 12 p.m. | 1 September at 12 p.m. | 0.87 kWh/m² ǀ 7.77 kWh |
| Summer solstice, 21st December 12 p.m. | 1 September at 12 p.m. | 0.65 kWh/m² ǀ 5.36 kWh |

The projected parametric model forms the remaining parametric part of the algorithmic–parametric modelling process, after the completion of the parametric optimization of the normalized top horizontal surface. The projected parametric model considers the same two parameters as the single parametric modelling process: the window central void amplitude and the circular modules thickness amplitude. Table 7 presents results of a single-objective optimization, considering these two parameters, of the divided and normalized cobogó with window, regarding insolation, points number with ideal illuminance (300–750 lux) and air temperature, for the solstices.

**Table 7.** Single-objective optimization results of the algorithmic–parametric model realized for the cobogó with window (type A), regarding insolation, points number with ideal illuminance and air temperature, for the winter and summer solstices.

| Single-Objective Optimization of the Algorithmic–Parametric Model | Parameter 1: Central Window Amplitude | Parameter 2: Circular Thickness Amplitude | Insolation (kWh ǀ kWh/m²) | Points Number with Ideal Illuminance (300–750 lux) ǀ Total Number of Points | Air Temperature (°C) |
|---|---|---|---|---|---|
| Winter solstice, 21st June 12 p.m. | 0.1 | 0.1 | 4.77 kWh ǀ 0.45 kWh/m² | - | - |
| | 0.1 | 0.3 | - | 86 ǀ 184 points | - |
| | 0.4 | 0.0 | - | - | 28.79 °C |
| Summer solstice, 21st December 12 p.m. | 0.1 | 0.1 | 3.37 kWh ǀ 0.32 kWh/m² | - | - |
| | 0.3 | 0.5 | - | 91 ǀ 184 points | - |
| | 0.1 | 0.0 | - | - | 33.49 °C |

For instance, the single-objective optimization of the insolation for the winter solstice presents a result of 4.77 kWh and 0.45 kWh/m², with 0.1 (0.13 m length and 0.12 m height) for parameter 1 and 0.1 (thickness of 0.134 m in bigger circles and of 0.108 m in smaller circles) for parameter 2. For the summer solstice, the single-objective optimization of the insolation has a result of 3.37 kWh and 0.32 kWh/m², with 0.1 (0.13 m length and 0.12 m height) for parameter 1 and 0.1 (thickness of 0.134 m in bigger circles and of 0.108 m in smaller circles) for parameter 2. Figure 12 displays a graphical representation with colours of the results of single-objective optimization processes, for winter and summer solstices.

We also developed a multi-objective optimization of the algorithmic–parametric model realized for the cobogó with window (type A), considering the same parameters that were considered in the parametric modelling process. The optimized shape solution considers simultaneously the performance obtained for insolation, points number with ideal illuminance and air temperature. Figure 13 displays the shape obtained for the divided and normalized shading cobogó with window, in the multi-objective optimization realized for the winter solstice day, illustrating with colour the insolation-optimized result.

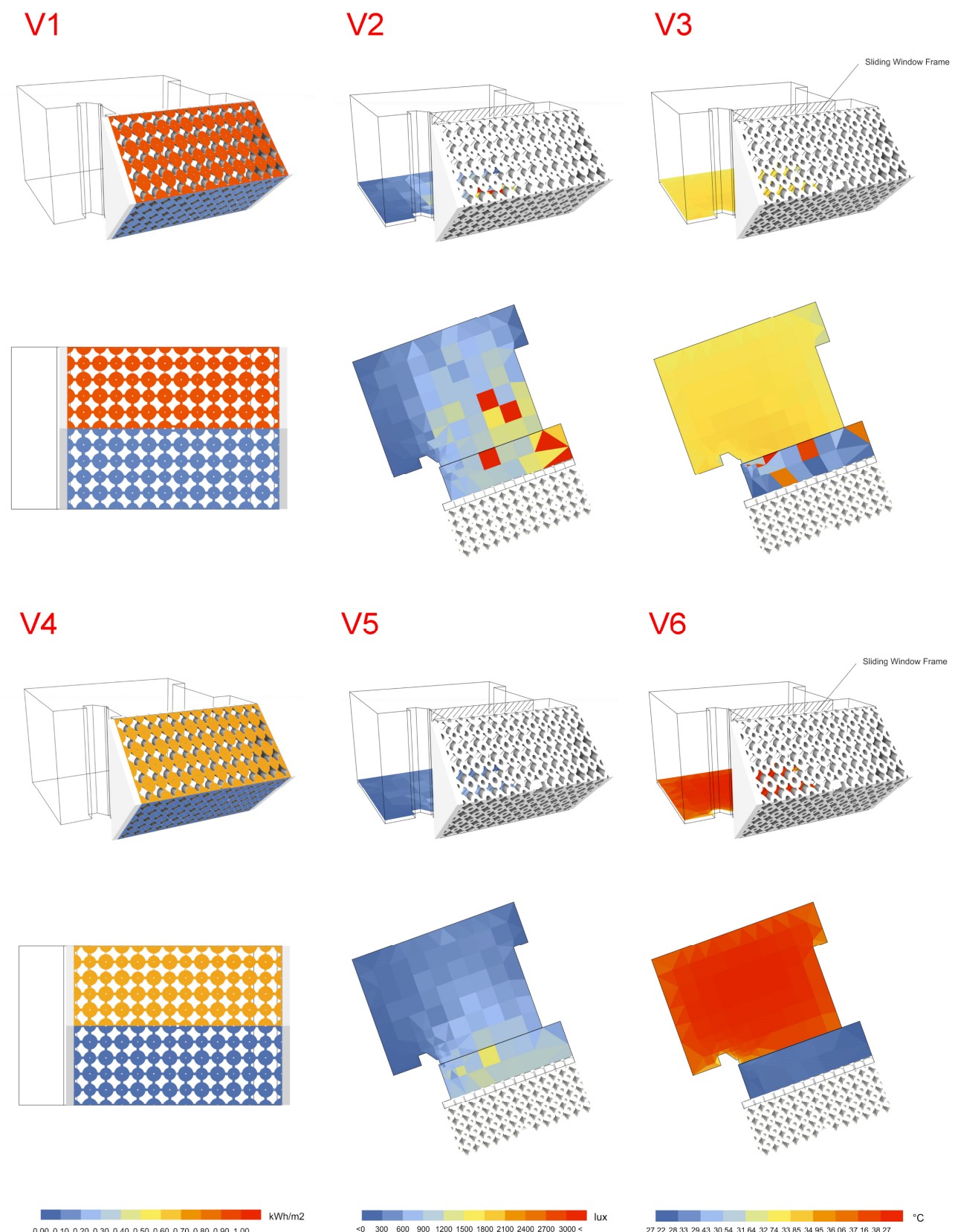

**Figure 12.** Single-objective optimizations results for winter solstice, June 21st 12 p.m. (images **V1**, **V2** and **V3**) and summer solstice, December 21st 12 p.m. (images **V4**, **V5** and **V6**). Results for insolation (**left** column), points number with ideal illuminance (**middle** column) and air temperature (**right** column).

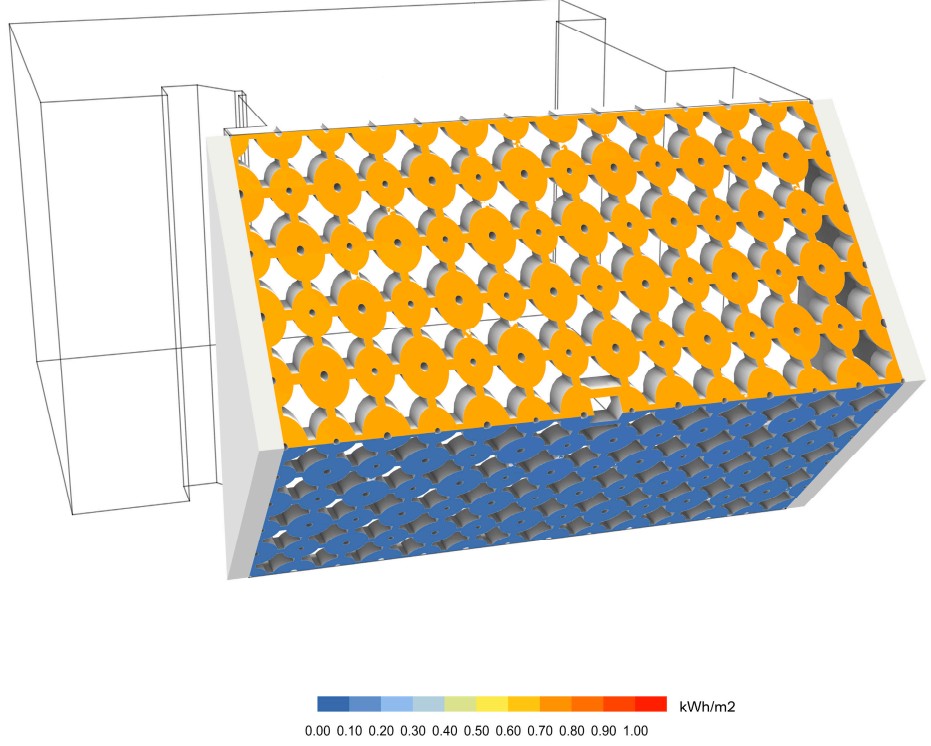

0.00 0.10 0.20 0.30 0.40 0.50 0.60 0.70 0.80 0.90 1.00   kWh/m2

**Figure 13.** Multi-objective optimization results, algorithmic–parametric cobogó with window, for insolation, points number with ideal illuminance and air temperature, in the summer solstice, 21st December 12 p.m.

Table 8 presents the results obtained in the multi-objective optimization realized for the algorithmic–parametric model of the cobogó with window (type A), regarding insolation, points number with ideal illuminance and air temperature, in the winter and summer solstices.

The winter solstice results are 4.70 kWh and 0.46 kWh/m$^2$ for insolation, 92 out of 184 possible points for ideal illuminance, 29.37 °C for temperature, 0.2 (0.26 m length and 0.24 m height) for parameter 1 and 0.3 (thickness 0.109 m in bigger circles and 0.089 m in smaller circles) for parameter 2. Summer solstice results are 3.59 kWh and 0.34 kWh/m$^2$ for insolation, 86 out of 184 possible points for ideal illuminance, 34.48 °C of air temperature, 0.2 (0.26 m length and 0.24 m height) for parameter 1 and 0.2 (thickness 0.122 m in bigger circles and 0.098 m in smaller circles) for parameter 2.

**Table 8.** Results of the multi-objective optimization realized for the algorithmic–parametric model of the cobogó with window, regarding insolation, points number with ideal illuminance and air temperature.

| Multi-Objective Optimization of the Algorithmic–Parametric Model | Parameter 1: Central Window Amplitude | Parameter 2: Circular Thickness Amplitude | Insolation (kWh | kWh/m$^2$) | Points Number with Ideal Illuminance (300–750 lux) | Total Number of Points | Air Temperature (°C) |
|---|---|---|---|---|---|
| Winter solstice, 21st June 12 p.m. | 0.2 | 0.3 | 4.70 kWh | 0.46 kWh/m$^2$ | 92 | 184 points | 29.37 °C |
| Summer solstice, 21st December 12 p.m. | 0.2 | 0.2 | 3.59 kWh | 0.34 kWh/m$^2$ | 86 | 184 points | 34.48 °C |

## 4. Discussion

This research explores computational methods as combinatorial, parametric and algorithmic–parametric modelling, together with single-objective and multi-objective optimization processes, regarding insolation, points number with ideal illuminance and air temperature, to improve the performance of the existing shading systems of the Nova Cintra building north façade.

Addressed in particular the module of the upper left corner of the north façade, analysing cobogó with window (type A) performance and exploring new shading possibilities.

Tables 9–11 compare the original shading solution with the generated solutions using the modelling and optimization processes. To compare the shading performance regarding insolation, points number with ideal illuminance and air temperature, we present the percentage difference in brackets. The generated solutions improved the performance for insolation, points number with ideal illuminance and air temperature. There are, however, some exceptions, when performance diminishes:

1. Insolation performance during the winter and summer solstices, using a combinatorial modelling with multi-objective optimization: insolation is 1.9 kWh (−16.4%) and 0.4 kWh (−15.4%);
2. Insolation performance during the winter solstice, using parametric modelling and single-objective optimization, algorithmic–parametric modelling and single-objective optimization, as well as algorithmic–parametric modelling and multi-objective optimization, where energy harvesting is 0.6 kWh/m² (−1.6%), 0.5 kWh/m² (−28.6%) and 0.5 kWh/m² (−27.0%), respectively;
3. Air temperature during the winter solstice, through algorithmic–parametric modelling and the multi-objective optimization process, where temperature increases for 29.4 °C (+0.6%), regarding the original shading.

The generated shading solutions performance achieve the best improvements regarding insolation, points number with ideal illuminance and air temperature, in the following cases:

1. Winter solstice, for insolation, with parametric modelling and single-objective optimization and with algorithmic–parametric modelling and single-objective optimization, where total insolation is 4.8 kWh (+111.1% than original shading). There is no energy harvesting efficiency improvement in these cases, the original shading efficiency value is 0.6 kWh/m², the same in combinatorial modelling and multi-objective optimization, as well as in parametric modelling and multi-objective optimization. In the summer solstice, the best improvements for insolation are in the algorithmic–parametric modelling and multi-objective optimization process, with a total insolation 3.6 kWh (+590.4%) and 0.3 kWh/m² (+240%) energy harvesting-efficiency;
2. For the points number with ideal illuminance, in winter solstice, the parametric modelling and multi-objective optimization process, obtains 106 points (+360.9% than the original shading). During the summer solstice, the best result is with parametric modelling and single-objective optimization process, with 118 points (+84.4%);
3. In the winter solstice, for air temperature, with parametric modelling and single-objective optimization, which reaches 28.3 °C (−2.9% than the original shading). During the summer solstice, with the algorithmic–parametric modelling and single-objective optimization, 33.5 °C (−3.0%) is obtained.

**Table 9.** Insolation in the original shading and the generated shading, winter and summer solstices.

| Insolation Analyses | Original Shading kWh \| kWh/m² | Combinatorial Modelling and Multi-Objective Optimization kWh \| kWh/m² | Parametric Modelling and Single-Objective Optimization kWh \| kWh/m² | Parametric Modelling and Multi-Objective Optimization kWh \| kWh/m² | Algorithmic-Parametric Modelling and Single-Objective Optimization kWh \| kWh/m² | Algorithmic-Parametric Modelling and Multi-Objective Optimization kWh \| kWh/m² |
|---|---|---|---|---|---|---|
| Winter solstice, 21st June 12 p.m. | 2.3 \| 0.6 | 1.9 \| 0.6 (−16.4% \| −) | 4.8 \| 0.6 (+111.1% \| −1.6%) | 4.7 \| 0.6 (+106.6% \| −) | 4.8 \| 0.5 (+111.1% \| −28.6%) | 4.7 \| 0.5 (+108.0% \| −27.0%) |
| Summer solstice, 21st December 12 p.m. | 0.5 \| 0.1 | 0.4 \| 0.2 (−15.4% \| +50%) | 1.1 \| 0.1 (+113.5% \| +40%) | 1.0 \| 0.1 (+98.1% \| −) | 3.4 \| 0.3 (+548.1% \| +220%) | 3.6 \| 0.3 (+590.4% \| +240%) |

**Table 10.** Points number with ideal illuminance (between 300 and 750 lux) in the original shading and the generated shading, in winter and summer solstices.

| Points with Ideal Illuminance (300–750 lux) Analyses | Original Shading Points \| Points | Combinatorial Modelling and Multi-Objective Optimization Points \| Points | Parametric Modelling and Single-Objective Optimization Points \| Points | Parametric Modelling and Multi-Objective Optimization Points \| Points | Algorithmic-Parametric Modelling and Single-Objective Optim Points \| Points | Algorithmic-Parametric Modelling and Multi-Objective Optimization Points \| Points |
|---|---|---|---|---|---|---|
| Winter solstice, 21st June 12 p.m. | 23 \| 178 | 28 \| 178 (+21.7%) | 95 \| 178 (+313.1%) | 106 \| 178 (+360.9%) | 86 \| 184 (+261.7%) | 92 \| 184 (+287.0%) |
| Summer solstice, 21st December 12 p.m. | 64 \| 178 | 112 \| 178 (+75%) | 118 \| 178 (+84.4%) | 96 \| 178 (+50%) | 91 \| 184 (+37.6%) | 86 \| 184 (+30.0%) |

**Table 11.** Air temperature (°C) in the original shading and the generated shading, in winter and summer solstices.

| Air Temperature Analyses | Original Shading °C | Combinatorial Modelling and Multi−Objective Optimization °C | Parametric Modelling and Single−Objective Optimization °C | Parametric Modelling and Multi−Objective Optimization °C | Algorithmic−Parametric Modelling and Single−Objective Optimization °C | Algorithmic−Parametric Modelling and Multi−Objective Optimization °C |
|---|---|---|---|---|---|---|
| Winter solstice, 21st June 12 p.m. | 29.2 | 28.4 (−2.9%) | 28.3 (−2.9%) | 29.1 (−0.4%) | 28.8 (−1.0%) | 29.4 (+0.6%) |
| Summer solstice, 21st December 12 p.m. | 34.5 | 33.6 (−2.6%) | 33.6 (−2.6%) | 33.6 (−2.6%) | 33.5 (−3.0%) | 34.5 (−0.1%) |

As the main findings, we highlight that the generated shading improvements in insolation are higher in the algorithmic–parametric modelling and respective optimization processes, in both solstices, especially in the summer solstice, the Sun's highest position, when the top normalized surface of the generated shading is further exposed to the Sun. In winter solstice, the sun lowest position, the improvement in the shading efficiency is considerable, but not as significant as in the summer solstice. The summer solstice has a greater improvement in the efficiency of harvesting sun energy, because the normalized top horizontal shading surface exposes more to the sunrays, comparing to the original shading surface.

Regarding the points number with ideal illuminance, the parametric modelling and multi-objective optimization process is more efficient during the winter solstice and the parametric modelling and single-objective optimization process has improved efficiency during the summer solstice. A hypothesis for the improved performance of the parametric modelling process, with single-objective and multi-objective optimizations, comparatively to the algorithmic–parametric modelling process, is that the parametric modelling process occurs in a vertical planar surface, allowing a homogeneous entry of natural daylight in the rooms. The algorithmic–parametric process generates two different shading surfaces, creating a distinct and heterogeneous daylight entry through these two surfaces. Future research intends to explore different parameterization processes in the algorithmic–parametric process: one parameterization process for the normalized top surface and another different parameterization process for the bottom-shading surface. This process expects to achieve

different daylight entries through the shading, for the top and bottom surfaces, to improve the efficiency in the points number with ideal illuminance inside the rooms.

For the air temperature, during the winter solstice, the parametric modelling and single-objective optimization process seems to be the most efficient, while for summer solstice the algorithmic–parametric modelling and single-objective optimization process have the best result. As well as in the case of the points number with ideal illuminance, there is the hypothesis that the number of shading surfaces and their orientation, one vertical surface in the parametric modelling process and two different surfaces, oriented in different ways, in the algorithmic–parametric process, influence the temperature performance. Therefore, future research for air temperature intends to verify also if the appliance of two different parameterization processes, for the two different surfaces of the algorithmic–parametric shading, allow improving shading performance.

Comparing the improvements obtained in this research with the improvements achieved by the researches described in the literature review, we can verify that Caldas and Norford [6] found an improvement of 15% between the worst and the best result achieved for the annual electrical energy consumption of an office building. These results are optimized with a genetic algorithm. The parametric space of solutions are the window dimensions in the façades of the building.

Jalali, Noorzai and Heidari [8], in the design of the overall form and façades windows of an office building, realized through parametric modelling, mention improvements of 8% (reduction in cooling load), 21% (reduction in heating load) and 37% (increase in Useful Daylight Illuminance). These improvements are also achieved with optimization algorithms, applied on the parametric space of solutions defined for the overall form and façades windows of the office building.

Fang and Cho [11], also using parametric modelling and optimization algorithms for the geometry definition of buildings and their façades, report improvements in energy savings of 30% in office buildings and of 1 to 3% in apartment buildings.

González and Fiorito [15], using the same processes for the optimization of external shadings in an office building, indicate improvements of 35% in energy savings and of 48% in the reduction of $CO_2$ emissions for the building.

Therefore, the improvements achieved in the reference researches, which are related as well with energy performance measures, are in a range between 1% and 48%.

In the present research the improvements, all of them obtained using optimization algorithms as well, are between 98.1% reached by parametric modelling and 590.4% attained by algorithmic–parametric modelling, for the insolation.

For the points number with ideal illuminance, the improvements are between 21.7%, obtained by combinatorial modelling, and 360.9%, reached by parametric modelling.

For air temperature, improvements are between 0.1%, achieved by algorithmic–parametric modelling, and 3.0%, attained also by algorithmic–parametric modelling.

Thus, the improvements obtained in this research are in general higher comparing with the improvements achieved by the researches described in the literature review, considering the values obtained for insolation and points number with ideal illuminance. In insolation, the best improvement values are achieved through algorithmic–parametric modelling, being considerably higher. In points number with ideal illuminance, the higher values are reached by parametric modelling. Regarding air temperature, the best improvement values obtained by algorithmic–parametric modelling are lower, being in the range achieved by the researches present at the literature review.

In general, algorithmic–parametric modelling demonstrated that can be more efficient than parametric modelling to obtain higher improvements in the energy performance of the studied shading system. Despite the improvements in points number with ideal illuminance are higher with parametric modelling, we believe that is possible to improve the algorithmic–parametric process, in order to achieve also for this energy measure better results than parametric modelling.

It should be referred also that combinatorial modelling has in general lower improvement results than parametric and algorithmic–parametric modelling types. This is due it having a narrow space of shape solutions. However, it has the advantage of being the quickest process of achieve improved optimized performance solutions for the shading system, of all the three tested modelling types in this research.

## 5. Conclusions

This research develops an energy-based design model, to improve the energy performance of a carioca modern building, the Nova Cintra. The research was conducted to improve the building performance using computational methods, such as combinatorial modelling, parametric modelling and algorithmic–parametric modelling, jointly with single-objective and multi-objective optimizations, applied over the existing shading systems on the building north façade. The proposal is an exploratory and prospective research, not intends at least in this stage to build new shading systems in a real context. The shading solutions are developments of the original shading systems, improving their performance regarding insolation, illuminance and air temperature. These improvements intend to avoid air-conditioning systems inside the building, as well as reduce the need of artificial lighting, diminishing the energy consumption. On the other hand, improvement in the performance regarding insolation explores the solar energy harvesting capacity in the building shading systems, which acquire the potential of generating renewable energy using photovoltaic technology.

The methodology for the first stage of the research modelled in 3D the Nova Cintra building using visual and textual programming, analysed in general all the building envelope and in detail their north façade, for insolation and illuminance, and explores also combinatorial modelling and multi-objective optimization processes, using the three original shading systems of the north façade. This improves the insolation and illuminance performance of the building façade.

The second research stage, for the upper left corner module of the north façade, generated solutions to improve shading, using parametric and algorithmic–parametric modelling with single-objective and multi-objective optimization processes, for insolation, illuminance, and air temperature.

Conclusions regarding the generated shading solutions performance:

1. The research use of modelling and optimization processes, namely the combinatorial, parametric and algorithmic–parametric modelling, with single and multi-objective optimizations, can improve the performance of the building's shading solutions, regarding insolation, illuminance and air temperature;
2. The algorithmic–parametric modelling, jointly with single-objective and multi-objective optimization processes, can improve better the insolation. The parametric modelling, with single-objective and multi-objective optimization, can improve better the illuminance. Both parametric (for winter solstice) and algorithmic–parametric (for summer solstice) modelling procedures, with single-objective optimization, achieved better performance for the air temperature;
3. The research developed delivers energy-based design to improve buildings' energy performance. Research confirms that shape transformation and modelling processes can improve energy performance.

This research performance improvement is in general higher than in previous researches described in the literature review. Algorithmic–parametric modelling in our research proved to be more efficient than parametric modelling and combinatorial modelling, improving the energy performance of the shading systems, especially regarding insolation.

This research has the following limitations: (1) the time expended on the optimization processes, using the Galapagos and Octopus plug-ins. Most of the optimizations took 24 to 48 h to be accomplished. (2) The period established for the realization of the research: 3 months. These aspects limited the realization of further algorithmic–parametric modelling

processes in obtaining higher performance. This research also chose not to apply the algorithmic–parametric modelling process to the entire north façade of the Nova Cintra building.

Future work should thus address the development of the algorithmic–parametric modelling type, looking to make it more efficient than parametric modelling to improve the number of points with ideal illuminance. Expecting to also increase the results already achieved for air temperature, it intends as well to develop a method to apply the study to the entire Nova Cintra north façade.

This research uses an energy-based design model to transform building-shading systems, using computational methods to improve the buildings energy performance for factors such as insolation, illuminance and air temperature. Energy-based design models can apply as well to design other building elements such as roofs and walls, or design entirely buildings. In the case of this research, the use of energy-based design in building-shading systems can minimize the building's energy consumption needs in devices such as air-conditioning systems and artificial lighting. Simultaneously, the improvements achieved in insolation can stimulate the harvesting of solar energy in the shading systems and the production of renewable energy. Using photovoltaic technology in the future, research might transform buildings beyond energy consumers to energy producers. The potential gain in the overall buildings' energy performance can diminish the use of energy from pollutant sources such as coal and increase the production and use of energy from renewable sources such as the Sun. This is a significant contribution, by human action, to mitigate climate change, a fundamental quest for the environmental sustainability of our planet.

**Author Contributions:** Conceptualization, D.M. and G.C.H.; methodology, D.M.; software, D.M. and G.C.H.; validation, D.M. and G.C.H.; formal analysis, D.M. and G.C.H.; investigation, D.M. and G.C.H.; resources, D.M. and G.C.H.; data curation, D.M. and G.C.H.; writing—original draft preparation, D.M.; writing—review and editing, D.M. and G.C.H.; visualization, D.M.; supervision, D.M. and G.C.H.; project administration, D.M. and G.C.H.; funding acquisition, G.C.H. All authors have read and agreed to the published version of the manuscript.

**Funding:** This research was funded by Brazilian scientific initiation grants and artistic and cultural initiation grants (PIBIC and PIBIAC) for Taiane Nepomuceno and Ana Clara Mouro (PIBIAC IC 2022 CLA/UFRJ Edital 449/2022), and Ronaldo Lee (PIBIAC IC 2021, CLA/UFR, Edital 212/2021) that we acknowledge and thank for their research contribution.

**Data Availability Statement:** Data sharing is not applicable to this article.

**Conflicts of Interest:** The authors declare no conflict of interest.

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
