# Peer review of "Energy-Based Design: Improving Modern Brazilian Buildings Performance through Their Shading Systems, the Nova Cintra Case Study"

_buildings, doi:10.3390/buildings13102543_

Round 1

Reviewer 1 Report

Comments and Suggestions for Authors

1. The abstract must present the primary information for the initial understanding of the article. There must be a balance of information to cover all sections adequately. The main findings were not explored. In addition, the objective of the work needs to be better presented.

2. In the Introduction section, the authors should better highlight the innovation of their research and what gaps in previous knowledge it covers.

3. In lines 743 and 744, the authors state, “This research develops an energy based design model to improve the energy performance of a carioca modern building, the Nova Cintra.” I recommend that a synthesis of the methodology be presented at the beginning of Materials and Methods Section. Also, is this methodology replicable? For what type of building? The study Implications need to be expanded.

4. The authors need to discuss the findings further. For example, in the researched literature, were other solutions with the same objective? If so, how does the proposed solution perform compared to these tools? There is a need to properly situate your findings in the context of the findings of other researchers using relevant references.

5. In the Conclusions Section, the authors should better highlight the relevance of the results, recommendations for future work, and limitations of the research.

Reviewer 2 Report

This manuscript presents an energy performance modeling framework  of the shading systems with both  combinatorial modeling and parametric modeling processes to transform the shading systems shape.  The work and results are overall well-designed and presented and I have only a few minor issues and comments:

  • The overall writing and grammar needs some polish work (third line should be ‘start by analyzing’, the abstract should also include quantitative description of the performance improvement, maybe extracts from Table 11)

  • The literature review section overall seems not well-structured. I think this section should be expanded with perhaps of 1, Why the shading system design is a critical task/issue. 2, What have been the methods deployed (combinatorial, parametric vs algorithmic parametric) and what are the main advantages (or disadvantages) 3. What are the challenges that this work is trying to overcome or value to add. 

  • I think the energy based model should be energy-based (this word has appeared many times)

  • In the Material/Methods section, many optimization models such as Octopus (line 397) show up, please add relevant citations and one or two sentences about what/why they are used.

The writing and grammar would need some proofreading

Reviewer 3 Report

The manuscript analyzes the energy performance of the shading systems of the Brazilian building Nova Cintra.

The study is very interesting, up-to-date for environmental issues and energy saving and of interest to the reader. However, a revision of the document structure and some additions are needed before publication:

introduction:

Review the general wording of the conclusions. There is a lack of framing and contextualisation of the problem. It is desirable to understand how the topic addressed is of interest to the building sector in the scenario of climate change and the need to reduce energy consumption in buildings. This is a study of shielding systems applied to the building envelope. Therefore, the concept of envelope as a filter between internal and external environment should be included (I recommend reading the paragraph 3.1 of the following publication: https://doi.org/10.1016/j.rser.2022.112850).

The theoretical background could be dealt with in a separate subsection or paragraph. The bibliographical references seem to be few.

Methodology:

The methodology is described clearly and in detail. It is recommended

Discussion and conclusions:

Clearly separate the contents of the two sections. In the conclusions, instead of a simple synthesis, clearly state the novelty of the present work compared to those mentioned in the theoretical background at the beginning of the article. And how can these screens help to mitigate the impact on the environment?

The figures and tables included in the text are clear and understandable.

Overall, an interesting work.

Round 2

Reviewer 1 Report

Dear authors,

Thank you for addressing my comments. The manuscript is significantly improved.

With kind regards,

Reviewer 3 Report

The authors have significantly improved the manuscript based on the suggestions. I think it can be published.